# Cellular differentiation into hyphae and spores in halophilic archaea

Shu-Kun Tang [1,17] ✉, Xiao-Yang Zhi [1,17], Yao Zhang[2,17], Kira S. Makarova [3,17], Bing-Bing Liu [1,4,17], Guo-Song Zheng[5], Zhen-Peng Zhang[2], Hua-Jun Zheng [6], Yuri I. Wolf [3], Yu-Rong Zhao[1], Song-Hao Jiang[2,7], Xi-Ming Chen[8], En-Yuan Li[1], Tao Zhang[2], Pei-Ru Chen[2,7], Yu-Zhou Feng[1], Ming-Xian Xiang[1], Zhi-Qian Lin[1], Jia-Hui Shi[2,7], Cheng Chang [2], Xue Zhang [4], Rui Li[1], Kai Lou[9], Yun Wang[9], Lei Chang[2], Min Yin[1], Ling-Ling Yang[1], Hui-Ying Gao[2], Zhong-Kai Zhang[10], Tian-Shen Tao[11], Tong-Wei Guan[12], Fu-Chu He[2], Yin-Hua Lu[5], Heng-Lin Cui[13], Eugene V. Koonin [3] ✉, Guo-Ping Zhao [14] ✉ & Ping Xu [2,7,11,15,16] ✉

Several groups of bacteria have complex life cycles involving cellular differentiation and multicellular structures. For example, actinobacteria of the genus *Streptomyces* form multicellular vegetative hyphae, aerial hyphae, and spores. However, similar life cycles have not yet been described for archaea. Here, we show that several haloarchaea of the family *Halobacteriaceae* display a life cycle resembling that of *Streptomyces* bacteria. Strain YIM 93972 (isolated from a salt marsh) undergoes cellular differentiation into mycelia and spores. Other closely related strains are also able to form mycelia, and comparative genomic analyses point to gene signatures (apparent gain or loss of certain genes) that are shared by members of this clade within the *Halobacteriaceae*. Genomic, transcriptomic and proteomic analyses of non-differentiating mutants suggest that a Cdc48-family ATPase might be involved in cellular differentiation in strain YIM 93972. Additionally, a gene encoding a putative oligopeptide transporter from YIM 93972 can restore the ability to form hyphae in a *Streptomyces coelicolor* mutant that carries a deletion in a homologous gene cluster (*bldKA-bldKE*), suggesting functional equivalence. We propose strain YIM 93972 as representative of a new species in a new genus within the family *Halobacteriaceae*, for which the name *Actinoarchaeum halophilum* gen. nov., sp. nov. is herewith proposed. Our demonstration of a complex life cycle in a group of haloarchaea adds a new dimension to our understanding of the biological diversity and environmental adaptation of archaea.

Archaea are a prokaryotic domain of life that structurally resemble, but are evolutionarily distinct, from bacteria. Apart from the sharp separation in the phylogenetic trees of universal genes (mostly encoding translation system components), archaea differ from bacteria in many major features including partly unrelated DNA replication and transcription machineries, different structures of membrane lipids and cell walls, and the corresponding, distinct enzymatic machineries involved in membrane and cell wall biogenesis,

several unique coenzymes, and unique RNA modifications[1]. A remarkable diversity of archaeal cell morphology matching that of bacterial cells has been discovered including lobe-shaped *Sulfolobus acidocaldarius*[2], filamentous *Methanospirillum hungatei*[3] and *Thermofilum pendens*[4], rod-shaped *Thermoproteus tenax*[5] and *Pyrobaculum aerophilum*[6], and even square-shaped *Haloquadratum walsbyi*[7]. The distinct shapes of these archaeal cells are maintained by the cell wall and cytoskeleton. However, despite the expanding research into archaeal biology including some limited cellular differentiation, such as cyst formation in *Methanosarcina*[8, 9], so far, there have been no reports of complex cellular differentiation accompanied by major morphological changes in archaea. This apparent lack of complex cellular differentiation in archaea contrasts the diverse different forms of cellular differentiation in bacteria that includes the formation of spores in bacilli and clostridia, heterocysts and akinetes in cyanobacteria, and particularly, complex differentiated colonies in myxobacteria and actinomycetes[10].

In this work, we describe a group of haloarchaea displaying cellular differentiation into hyphae and spores, isolated from salt marsh sediment. The morphogenesis of this haloarchaea visually resembles that of *Streptomyces* bacteria. Comparative genomics suggests potential gain and loss of genes that might be relevant to this type of archaeal morphogenesis. A gene encoding a Cdc48-family ATPase, and a gene cluster encoding a putative oligopeptide transporter, might be involved in cellular differentiation, as suggested by multi-omic analyses of non-differentiated mutants. Remarkably, this gene cluster from haloarchaea can restore hyphae formation of a *Streptomyces coelicolor* mutant that carries a deletion in a homologous gene cluster (*bldKA-bldKE*). These results provide new knowledge on the morphology of archaea and enrich our understanding of the biological diversity and environmental adaptation of archaea.

## Results and discussion

### A new haloarchaeal lineage with complex morphological differentiation

During the investigation of actinobacterial diversity from hypersaline environments, an unexpected hyper-halophilic archaeon with a distinct growth pattern was discovered serendipitously from the Qijiaojing Salt Lake in the Xinjiang Uygur Autonomous Region of China. This novel strain, designated YIM 93972, was isolated from a soil sample using the standard dilution-plating technique at 37 °C for four weeks on Modified Gause (MG) agar plates supplemented with 20% NaCl (w/v), which previously has been mainly used to isolate halophilic actinomycetes[11]. The colonies of YIM 93972 comprised yellow, orange, and red branching filaments with white aerial hyphae, underwent complex cellular differentiation (Fig. 1a, Supplementary Fig. 1a), and produced spores (0.5–0.7 × 1.0–1.2 μm in size) in both solid (Fig. 1b I-III) and liquid media (Fig. 1b IV). As other spore-forming strains[12], the spores of this new halophilic archaeon were resistant to heat stress up to 70 °C. In contrast, substrate hyphae (SH) were resistant up to 60 °C (Supplementary Fig. 1b). The higher resistance to heat stress of spores might enable them to survive under extreme saline conditions by their protective and repair capabilities. Optimal growth of YIM 93972 was observed at 40–45 °C, 3.8–4.3 M NaCl and 7.0–7.5 pH (Supplementary Data 1). Transmission electron microscopy (TEM) revealed the different stages of spore differentiation, from cell constriction, septa formation, and rod separation to chain formation (Fig. 1b V and VI). Thus, the cellular differentiation of this new haloarchaeon includes the growth of substrate and aerial hyphae and spore formation, which resembles the morphogenetic development of *Streptomyces*.

Analysis of the 16S rRNA gene indicated that YIM 93972 was closely related to *Halocatena pleomorpha* SPP-AMP-1[T], with 96.65% identity, and the two species are likely to represent a distinct lineage within

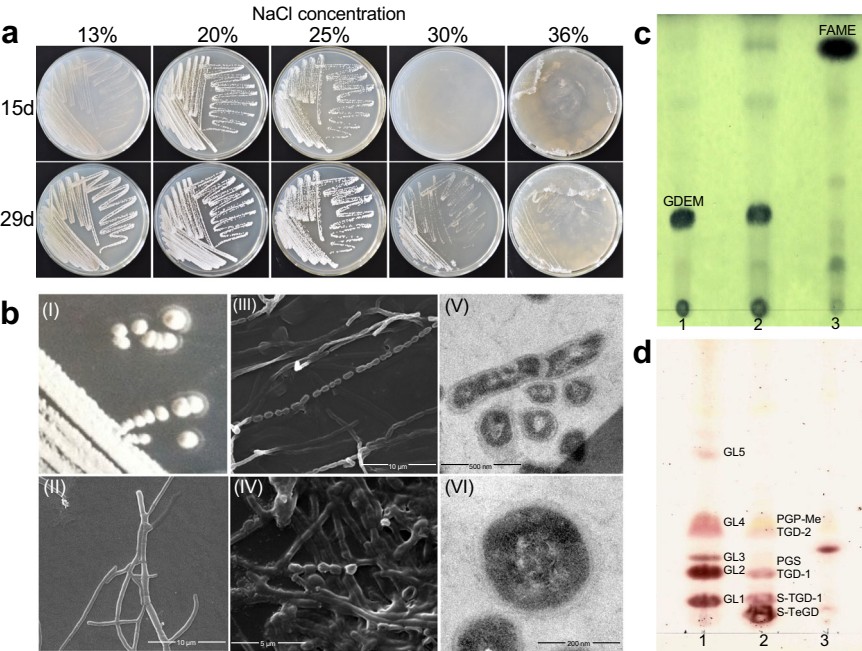

**Fig. 1 | Characteristics of haloarchaeal strain YIM 93972. a** Salt tolerance of strain YIM 93972 grown in ISP 4 medium after 15 and 29 days (*n* = 3). **b** Colony morphology (I) and scanning electron micrographs (EM) of strain YIM 93972 cultivated on solid and liquid ISP 4 medium containing 25% NaCl (*n* = 3). Scanning EM images of substrate mycelia (II; bar, 10 μm), aerial mycelia (III; bar, 10 μm), and long rod-like spore chains (III; spore size, 0.5–0.7 × 1.0–1.2 μm). Scanning EM images of mycelia grown in liquid ISP 4 medium (IV; bar, 5 μm). Transmission EM images (V and VI) of cells collected from ISP 4 plate after sporulation (bar, 200 nm). **c** Thin-layer chromatographic analysis of whole-organism methanolysates. Lanes: 1, *Haloferax volcanii* CGMCC 1.2150[T]; 2, YIM 93972; 3, *Escherichia coli* K12. **d** Analysis of the polar lipid composition of YIM 93972 using one-dimensional thin layer chromatography. Lanes: 1, YIM 93972; 2, *Halomarina oriensis* JCM 16495[T]; 3, *Halomarina salina* ZS-57-S[T]. The origin is at the bottom. Abbreviations: FAMEs, fatty acid methyl esters; GDEMs, glycerol diether moieties; GLs, glycolipids; PGP-Me, phosphatidylglycerol phosphate methyl ester; PGS, phosphatidylglycero-sulfate; TGD-1, galactosyl mannosyl glucosyl diether; TGD-2, glucosyl mannosyl glucosyl diether; S-TeGD, sulfated galactosyl mannosyl galactofuranosyl glucosyl diether; S-TGD-1, sulfated galactosyl mannosyl glucosyl diether.

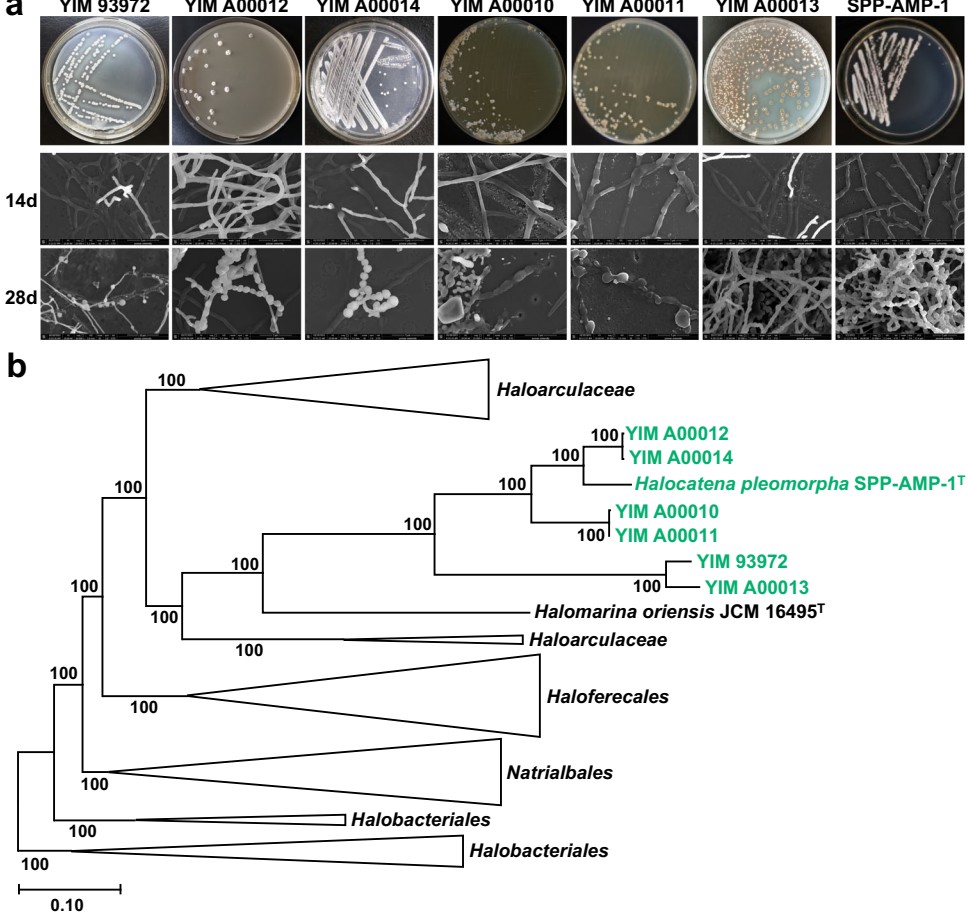

**Fig. 2 | Morphology and phylogeny of morphogenetic haloarchaea. a** Colony morphologies (first row) and scanning electron micrographs (second and third rows) of 7 morphogenetic *Halobacteria* grown in ISP 4 medium after 14 and 28 days ($n = 3$). **b**, Phylogenetic placement of morphogenetic isolates (green) within *Halobacteria*. The Maximum Likelihood phylogenetic tree (IQ-Tree[59], LG + F + R10 model) is based on 268 orthologous genes that are universal among 130 *Halobacteria* genomes and have at most 4 additional paralogs in any of these. The tree is rooted at the *Halobacterium-Halodesulfurarchaeum* clade following[60]. Branch support values were estimated using aBayes. Scale, substitutions per amino acid site.

the family *Halobacteriaceae*. Although YIM 93972 displays morphological features resembling those of actinomycetes, it is otherwise a typical archaeon that, in particular, contains glycerol diether moieties (GDEMs) and phosphatidylglycerol phosphate methyl ester (PGP-Me) in membrane lipids and an S-layer cell wall (Fig. 1c, d, Supplementary Fig. 2, Supplementary Data 1). In addition, genome analysis and proteomic data indicate that, like many extreme halophilic archaea, YIM 93972 produces halorhodopsin (Supplementary Fig. 3), a 7-transmembrane protein that function as a light-driven ion transporter. However, YIM 93972 lacks the genes encoding enzymes involved in the biosynthesis of bacterioruberin (in particular, lycopene elongase, halo.01227), a $C_{50}$ carotenoid that has been identified in several extremely halophilic archaea, the production of which is regulated by halorhodopsin[13, 14].

To investigate the ecological distribution of YIM 93972 and its relatives, we collected 15 soil samples from the Aiding Lake, Large South Lake, Dabancheng East Lake, and Uzun Brac Lake of Xinjiang Uygur Autonomous Region and amplified the 16S rRNA genes from environmental DNAs using a pair of PCR primers designed based on the 16S rRNA gene sequence of YIM 93972. In two soil samples from Aiding Salt Lake, 47 sequences of 16S rRNA gene were obtained and shown to form a clade together with YIM 93972 (OTU13 in Supplementary Fig. 4a, sequences in Supplementary Data 18). Thus, YIM 93972 and related haloarchaea are widespread in the Salt Lake

environment. Additionally, an attempt was made to isolate archaea related to YIM 93972 from the same soil samples from these Salt Lakes using the same method on Modified Gause agar plates. Finally, three strains YIM A00010, YIM A00011, and YIM A00014 from the Aiding Salt Lake as well as two strains YIM A00012 and YIM A00013 from the Uzun Brac Salt Lake were confirmed as novel archaeal strains with morphological differentiation (Fig. 2a).

The phylogenetic tree of 16S rRNA showed that *H. pleomorpha* SPP-AMP-1[T] isolated recently from a man-made saltpan site in India[15] and *Halomarina oriensis* JCM 16495[T] isolated from a seawater aquarium in Japan[16] were the two known species most closely related to the six new strains with morphological differentiation (Supplementary Fig. 4b). Cells of *H. oriensis* JCM 16495[T] were irregular coccoid or discoid shapes, but those of *H. pleomorpha* SPP-AMP-1[T] were pleomorphic in addition to rod-shaped. These two strains and the other two pleomorphic strains (*Haloferax volcanii* CGMCC 1.2150[T] [17] and *Haloplanus salinarum* JCM 31424[T] [18]) were collected for morphological comparison. On the same ISP 4 medium containing 20% NaCl, only *H. pleomorpha* SPP-AMP-1[T] exhibited differentiated morphology similar to that of YIM 93972 (Fig. 2a, Supplementary Fig. 5 and 6). Collectively, these observations demonstrated that a distinct group of morphologically differentiating (hereafter morphogenetic, for brevity) haloarchaea is widespread in various hypersaline environments.

## Genomic features of morphogenetic haloarchaea

To gain further insight into the biology of morphogenetic haloarchaea (referring specifically to haloarchaea with hyphae differentiation and spore formation), the complete genomes of YIM 93972 and other five morphogenetic strains were sequenced. The genome of YIM 93972 is composed of a major chromosome (2,676,592 bp, 57.2% G + C content, 2803 predicted genes), a minor chromosome (844,905 bp, 54.7% G + C content, 722 predicted genes), and three plasmids (Supplementary Fig. 7; Supplementary Data 2a). Altogether, the genome of YIM 93972 encompassed 3744 predicted protein-coding genes, 47 tRNAs, and two rRNA operons (the two 16S rRNA genes shared 99.8% identity) located on the major and minor chromosomes. Although all six genomes of the morphogenetic haloarchaeal strains consisted of two chromosomes of different sizes, otherwise, we identified many distinct genomic features (Supplementary Data 2, Supplementary Data 2b–f). In particular, the genome sizes of YIM A00010 and YIM A00011 (5.97 Mb and 5.82 Mb, respectively) are substantially larger than those of the other four strains (3.23 Mb on average). Among the six strains, only YIM A00013 had two rRNA operons like YIM 93972, but the genome of the former carried fewer plasmids and encompassed fewer genes than the latter.

We then constructed a set of 14,870 clusters of haloarchaeal orthologous genes (haloCOGs; Supplementary_data_file_1). Phylogenomic analysis of 268 low-paralogy haloCOGs that are universally conserved in 130 genomes of class *Halobacteria* (Supplementary Data 3) showed that YIM 93972 and its morphogenetic relatives including *H. pleomorpha*, formed a clade with *Halomarina oriensis*, within the family *Halobacteriaceae* (Fig. 2b, Supplementary_data_file_2). The morphogenetic haloarchaea shared a core of 2,053 strictly conserved orthologous genes, but lacked 336 genes that are common in other *Halobacteria* (Supplementary Data 3, Supplementary Data 4a). For 3208 of the 3744 predicted genes of YIM 93972, an ortholog was identified in at least one non-morphogenetic haloarchaeon. After mapping the patterns of gene presence-absence in haloCOGs onto the species tree, we obtained a maximum likelihood reconstruction of the history of gene gains and losses in class *Halobacteria*. The ancestor of the morphogenetic haloarchaea was estimated to contain 3903 haloCOGs, of which 1420 were represented in the tree branch that separates the morphogenetic clade from the common ancestor with the sister clade, whereas 339 new genes were gained (Supplementary Data 4b).

Several gene losses appeared to be relevant for the morphogenesis of these new haloarchaea, including the hyphae formation, and lack of cell motility (Supplementary Data 4c). These include the loss of CetZ (FtsZ3) which is involved in controlling the rod-like cell shape in non-morphogenetic haloarchaea[19], the archaellum and another halobacterial type IV pili system, typified by the PilB3 locus of *Haloferax volcanii*[20], which is represented in *Halobacteria* and *Methanomicrobia*[20]. Rod-shaped cells were not observed in morphogenetic haloarchaea, suggesting a functional link between the loss of *cetZ* and complex hyphae differentiation. Conversely, in morphogenetic haloarchaea, the halo.02163 family of MreB-like proteins is expanded, whereas the halo.02581 family of MreB-like proteins that is common in other haloarchaea is lost. Given that MreB is the key cell shape determinant in bacteria[21], these findings further emphasize specific changes in cytoskeleton organization that are likely to be relevant for the pleomorphism of these archaea (Supplementary Fig. 8).

Among the gains, there are 61 genes that are strictly conserved in all 7 morphogenetic haloarchaea and absent in other *Halobacteria* in our genome set. These gained genes included a distinct homolog of Cdc48, an AAA+ ATPase (halo.06695, *ORF_0238*). The proteins of this superfamily play key roles in a variety of cellular processes, including cell-cycle regulation, DNA replication, cell division, the ubiquitin-proteasome-system and others[22, 23] and might contribute to the morphological differentiation. Another notable case is a M6 family metalloprotease (halo.06530 and halo.06177, *ORF_0582* and *ORF_1236*), a distant homolog of InhA1, a major component of the exosporium in *Bacillus* spores[24]. In most archaea, the S-layer is the exclusive cell envelope component that plays a crucial role in surface recognition and cell shape maintenance[25, 26]. The morphogenetic haloarchaea shared a distinct of S-layer-forming glycoprotein (halo.06692, *ORF_1400*), which might be related to the unique morphology. Furthermore, proteomic data showed that YIM 93972 produced two varieties of S-layer proteins encoded by *ORF_1400* and *ORF_1704* (Supplementary Fig. 9a–d). For most of the remaining genes gained and, in some cases, duplicated in the morphogenetic haloarchaea, there was no functional prediction or only a generic prediction, although notably many of these genes encoded predicted transmembrane or secreted proteins (Supplementary Data 4b, d). Several of these proteins contain aspartate-rich repeats that could coordinate $Ca^{2+}$ ions, which are known to activate spore germination in sporulating actinobacteria[27].

Sporulating bacteria employ a distinct set of small molecules for energy storage and spore protection. We identified genes present in all 7 genomes of morphogenetic haloarchaea that are involved in trehalose biosynthesis and utilization[28], dipicolinic acid biosynthesis[29] and poly(R)-hydroxyalkanoic acid biosynthesis[30] (Supplementary Data 3). Comparative omics data indicated that genes involved in the biosynthesis of small molecules implicated in sporulation, namely, poly(R)-hydroxyalkanoic acid synthesis (*ORF_2608*) and 4-hydroxy-tetrahydrodipicolinate (*ORF_1392*), were upregulated 3.6 and 7.8 folds at the mRNA, and 1.3 and 1.6-fold at the protein level, respectively. This observation suggests that poly(R)-hydroxyalkanoic acid could be a key storage molecule in the spores. Polyhydroxyalkanoates also accumulate in hyphae and spores in some actinobacterial species[30]. However, all these genes are also widespread in non-morphogenetic *Halobacteria*. Another notable gene module present in all genomes of morphogenetic haloarchaea consists of a MoxR family ATPase and a von Willebrand factor type A (vWA) domain containing protein, which is strongly associated with bacterial multicellularity[31].

## Random mutagenesis and comparative multi-omics uncover several morphogenetic genes

Seeking to identify the genetic basis of cellular differentiation in morphogenetic haloarchaea, we performed nitroso-guanidine (NTG) chemical mutagenesis on spores of YIM 93972. After four generations of serial cultivation from the original 1110 mutant colonies, we obtained five transitional (weak aerial hyphae) and three bald (no aerial hyphae) mutants (Supplementary Fig. 10a, b). Analysis of the genome sequences of these 8 mutants identified two mutations in protein-coding gene sequences and one mutation upstream of a gene in all three bald mutants; and one mutation upstream of a gene in three of the five transitional mutants (Supplementary Fig. 10c, Supplementary Data 5). In the three bald mutants, the two intragenic mutations were the non-synonymous G1067A substitution in *ORF_0238* (ATPase of the AAA+ class, Cdc48 family), and synonymous C309T substitution in *ORF_0964* (RNA methyltransferase, SPOUT superfamily), whereas the third mutation was the T136G substitution in the non-coding region upstream of *ORF_2797* (ParB-like DNA binding protein). In five transitional mutants, the mutation shared by three mutants (T2, T4, and T5) was C69T in the intergenic region upstream of *ORF_1717* (Uncharacterized protein).

Given the lack of an experimental genetic system for YIM 93972, we performed whole transcriptomic (Fig. 3a, Supplementary Data 6–9) and quantitative proteomic analyses (Fig. 4a, Supplementary Fig. 11, Supplementary Data 10–12) for two transitional and three bald mutants to further characterize the potential association of the identified mutations with the mutant phenotypes. We identified the G1067A substitution in *ORF_0238* in the transcriptome of all

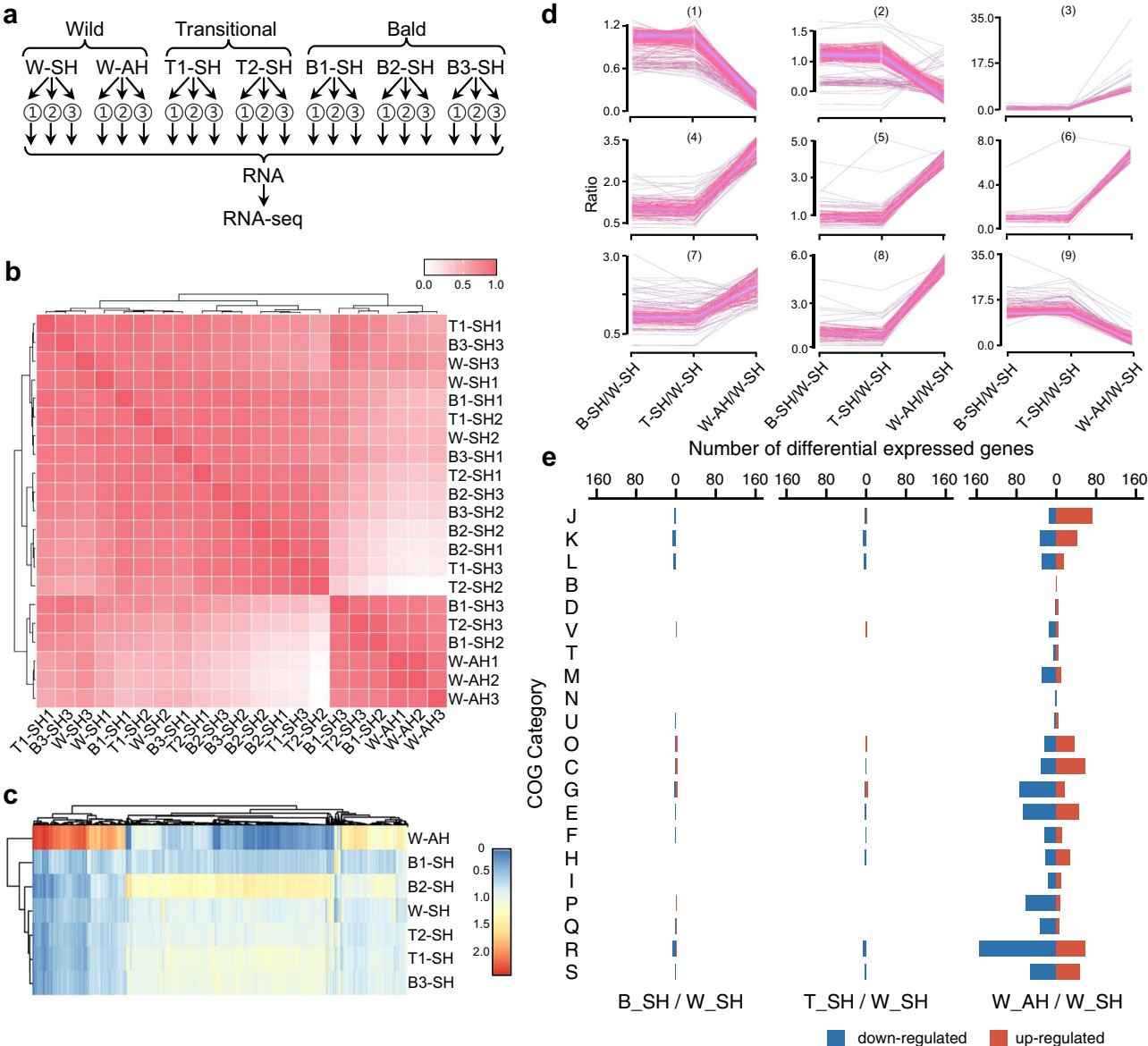

**Fig. 3 | Transcriptomic analysis of strain YIM 93972 wild type and non-differentiating mutants. a** Schematic workflow for transcriptomic analysis. We collected cultures representative for aerial hyphae (AH) and substrate hyphae (SH) from wild-type colony, and the morphological mutants. Two transitional (T) and three bald (B) colonies were chosen for the mutants. Each sample included 3 technical repetitions. Total 16 culture plates were randomly pooled for W-SH and W-AH sample, respectively. Total 32 culture plates were randomly pooled for each mutant sample. **b** Spearman's correlation coefficients for transcriptomic profiling of 21 samples. The $x$ and $y$ axes represent the $\log_2$-transformed gene intensities in each two-sample comparison. **c** Heat maps of differentially expressed genes. Red indicates upregulated genes; blue, downregulated genes. Statistical differences between two groups were analyzed using two-tailed unpaired $t$-tests. $P < 0.05$ was considered statistically significant. **d** The expression clusters of total differentially expressed genes. All dysregulated genes were clustered into different expression groups according to the fold change based on the gene expression value at the

mRNA level in different samples by Mfuzz (v2.58.0)[75]. **e** Functional classification of dysregulated genes according to COG functional categories. The COG categories are listed as follows. J: Translation, ribosomal structure and biogenesis; K: Transcription; L: Replication, recombination and repair; B: Chromatin structure and dynamics; D: Cell cycle control, cell division, chromosome partitioning; V: Defense mechanisms; T: Signal transduction mechanisms; M: Cell wall/membrane/envelope biogenesis; N: Cell motility; U: Intracellular trafficking, secretion, and vesicular transport; O: Posttranslational modification, protein turnover, chaperones; C: Energy production and conversion; G: Carbohydrate transport and metabolism; E: Amino acid transport and metabolism; F: Nucleotide transport and metabolism; H: Coenzyme transport and metabolism; I: Lipid transport and metabolism; P: Inorganic ion transport and metabolism; Q: Secondary metabolites biosynthesis, transport and catabolism; R: General function prediction only; S: Function unknown.

transitional mutants (except for two biological replicates of mutant T2) and bald mutants but not the wild-type transcriptome (Supplementary Fig. 10d, Supplementary Data 7). Furthermore, *ORF_0238* was downregulated 3.9 fold ($p < 0.001$) at the protein level in the bald mutants (Supplementary Data 11-12). Phylogenetic analysis of the Cdc48 family indicated that *ORF_0238* belonged to a distinct branch specific for morphogenetic haloarchaea (Supplementary Fig. 12).

Taken together, these observations suggest that *ORF_0238* might be functionally important for sporulation and morphological differentiation in YIM 93972.

To further characterize the genetic basis of hyphal differentiation, we compared the transcriptome and quantitative proteome of the wild type from both aerial hyphae and substrate hyphae with those of morphogenetic mutants from substrate hyphae only (given the

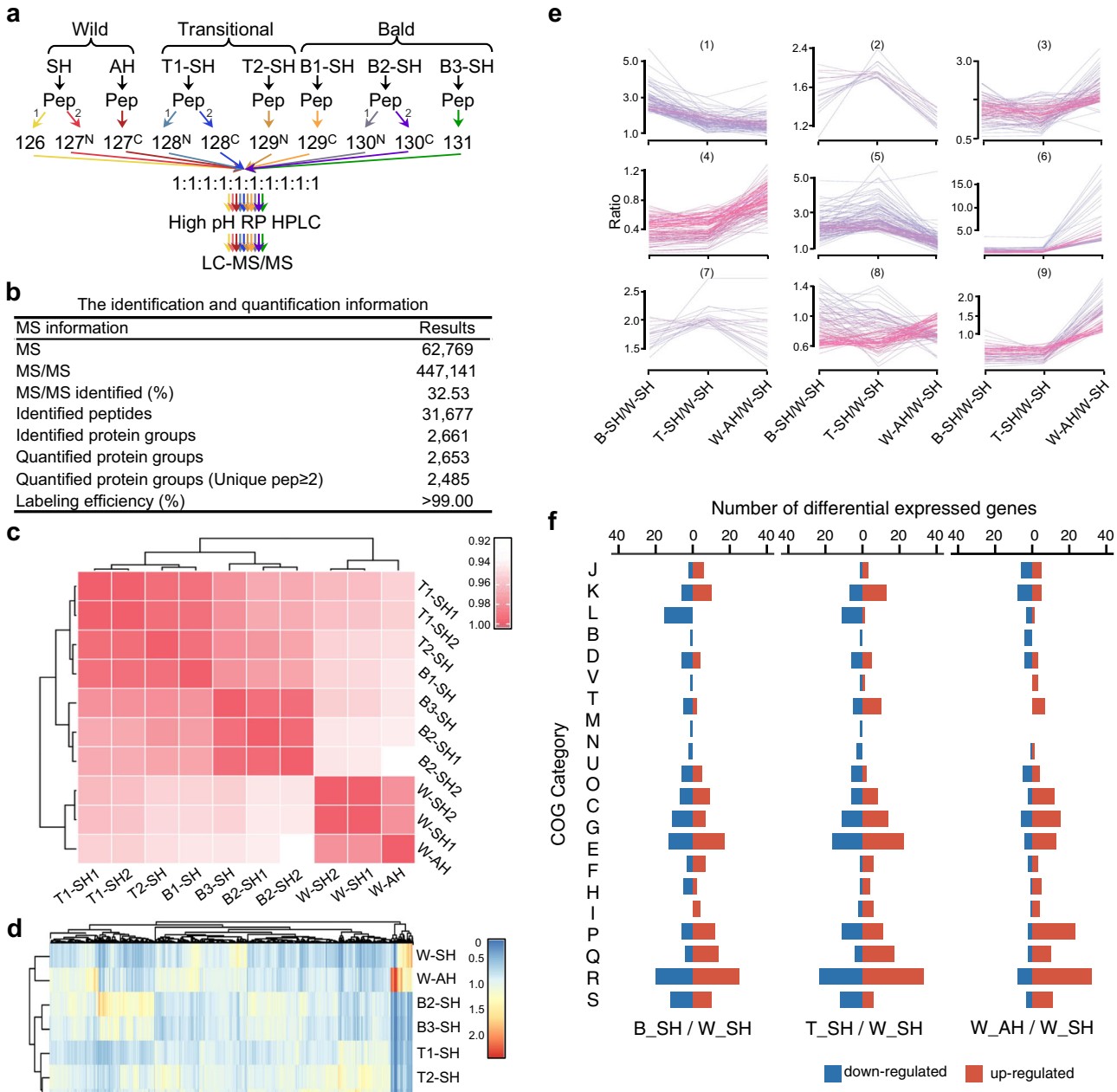

**Fig. 4 | Proteomic analysis of strain YIM 93972 wild type and non-differentiating mutants. a** Overview of TMT methodology for multiplexed comparative analysis. Abbreviations: AH, aerial hyphae; SH, substrate hyphae; W, wild type; T, transitional mutant; B, bald mutant. Two transitional and three bald colonies were selected for the mutants. Each group included two technical repetitions, W-SH1 and W-SH2 in wild group, T1-SH1 and T1-SH2 in transitional group, and B2-SH1 and B2-SH2 in bald group, respectively. Total 16 culture plates were randomly pooled for W-SH and W-AH sample, respectively. Total 32 culture plates were randomly pooled for each mutant sample. **b** Summary of MS identification and quantitation. **c** Spearman's correlation coefficients for proteome profiling of ten

samples. The *x* and *y* axes represent the log$_2$-transformed protein intensities in each two-sample comparison. **d** Heat maps of differentially expressed proteins. Red indicates upregulated proteins; blue, downregulated proteins. Statistical differences between two groups were analyzed using two-tailed unpaired Significance A[70]. $P < 0.05$ was considered statistically significant. **e** The expression clusters of total differentially expressed proteins. All dysregulated proteins were clustered into different groups according to the fold change based on the protein expression value in different samples by Mfuzz (v2.58.0)[75]. **f** Functional classification of dysregulated genes according to COG functional categories. The annotations of COG categories are listed in the legend of Fig. 3.

absence of aerial hyphae) (Figs. 3a, b and 4a–c, Supplementary Fig. 11a–e). Quantitative analysis revealed 107 genes (Fig. 3c–d, Supplementary Data 9, Supplementary Data 19) and 118 proteins (Fig. 4d, e, Supplementary Data 12, Supplementary Data 20) that were significantly upregulated in the aerial hyphae of wild type compared to the substrate hyphae of both wild and mutant types. The differentially expressed genes and proteins play major roles in translation, transcription, metabolism, and ion transport (Figs. 3e and 4f,

Supplementary Data 13). In particular, ATP-binding cassette (ABC) peptide transporter operon (*ORF_2669-ORF_2673* on chromosome I) was significantly upregulated in wild-type aerial hyphae (Fig. 5a, b, Supplementary Data 14). In this operon, the peptide-binding protein (*ORF_2669*) was significantly upregulated 6.91 and 1.6 fold at the mRNA and protein levels, respectively (Fig. 5c, Supplementary Data 21).

It has been shown that *Streptomyces coelicolor* employs a distinct ATP-binding cassette (ABC) transporter to import oligopeptides that

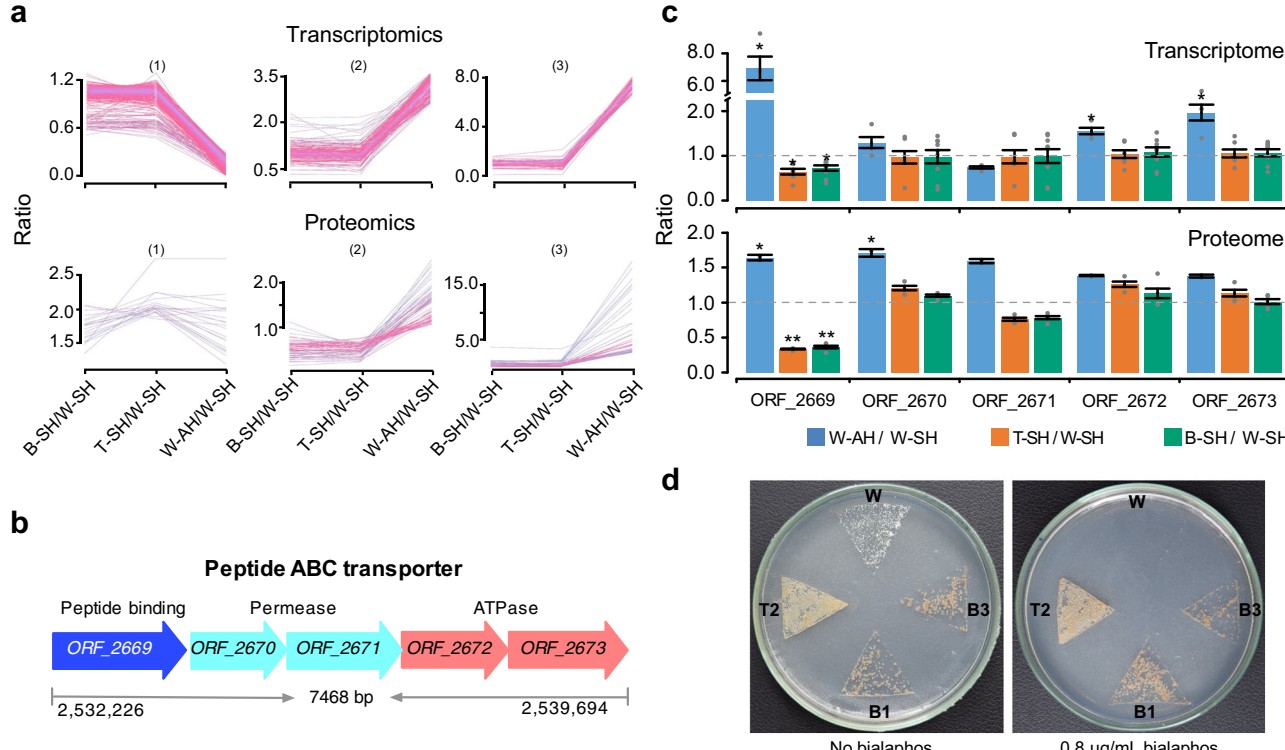

**Fig. 5 | Sporulation-related genes of strain YIM 93972. a** Typical gene expression profiles in varied differentiating conditions at the mRNA and protein levels. Abbreviations: AH, aerial hyphae; SH, substrate hyphae; W, wild type; T, transitional mutant; B, bald mutant. **b** Gene organization of the peptide ABC transporter in strain YIM 93972. Arrows indicate the size and directions of ORFs. **c** Expression levels of the peptide ABC transporter encoding genes at the mRNA and protein level. In transcriptomic data, gene expression at the mRNA level between two groups was calculated by TPM value and shown as means±MSD from W-AH/W-SH (*n* = 3), T-SH/W-SH (*n* = 6), and B-SH/W-SH (*n* = 9). In proteomic data, gene

expression at the protein level between two groups was calculated by intensity and shown as means ± MSD from W-AH/W-SH (*n* = 2), T-SH/W-SH (*n* = 3), and B-SH/W-SH (*n* = 4). Statistical differences between two groups were calculated using two-tailed unpaired *t*-tests (inter-group) at the mRNA level and Significance A[70] (intra-group) at the protein level, respectively. *P < 0.05; **P < 0.01; ***P < 0.001. The fold change and *p*-Value were shown in Supplementary Data 4 for Fig. 5c. **d** The morphological mutants are resistant to the toxic peptide bialaphos (*n* = 3). The strains listed above were grown for 14 days on ISP 4 agar plate containing 0 (left plate) and 0.8 (right plate) μg/mL bialaphos.

serve as signals for aerial mycelium formation[32]. To test whether YIM 93972 similarly used ABC transporters for importing peptides into the mycelia, we tested the resistance of transitional and bald mutants to bialaphos, an antibiotic that enters cells via oligopeptide permeases[33]. We found that only the wild type was sensitive to bialaphos (Fig. 5d) whereas all mutants were resistant to this antibiotic up to concentrations less than 1.0 μg/mL (Supplementary Fig. 13). Despite the low protein sequence similarity with the *S. coelicolor* oligopeptide permease (Fig. 6a), we hypothesized that the *ORF_2669-ORF_2673* operon of YIM 93972 is involved in the aerial mycelium and spore formation. Deletion of the oligopeptide-transport operon (*bldKA-KE*) caused a bald phenotype (only present substrate mycelium) in *S. coelicolor*[34]. We introduced the entire *ORF_2669-ORF_2673* operon of YIM 93972 into the *S. coelicolor* M145 Δ*bldKA-KE*/pIB139 mutant (Supplementary Data 15, 16) and found that aerial mycelium formation and sporulation were restored (Fig. 6b). Thus, the *ORF_2669-ORF_2673* operon of YIM 93972 might be involved in aerial mycelium formation by importing signaling oligopeptides, and could be functionally equivalent to the *bldKA-KE* operon of *S. coelicolor*.

In Actinobacteria, Cyanobacteria and the sporulating Firmicutes, multiple transcriptional regulators (TRs) are involved in complex cell differentiation[35–38]. In our transcriptomic and proteomic data, 47 TRs were found to be significantly up-or down-regulated in aerial hyphae comparted to substrate hyphae (Supplementary Data 17). These included four TRs regulating phosphate uptake (*ORF_0890, ORF_0891,* and *ORF_1046*) as well as *ORF_1998*, the cell fate regulator YlbF (YheA/YmcA/DUF963 family), which were all significantly upregulated. The

ortholog of *ORF_1998* is involved in competence development and sporulation in *Bacillus subtilis*[39]. Another upregulated gene, *ORF_1932*, encodes the global TR BolA, which is a key player in biofilm development[39–41]. Furthermore, the AbrB family TRs encoded in the same operon with genes involved in poly(R)-hydroxyalkanoic acid biosynthesis, the energy storage molecule in spores, was also upregulated. These findings are compatible with the major roles of distinct TRs in the cellular differentiation of morphogenetic archaea.

In summary, we report the biological, genetic and biochemical characteristics of a haloarchaeon that displays complex cellular differentiation. Although the mechanism of this differentiation remains to be elucidated, our findings implicate several genes, in particular, a distinct Cdc48-like ATPase that was mutated in all non-differentiating mutants. The observation that the morphogenetic haloarchaea are closely related to each other and, in the phylogenetic tree of *Halobacteria*, form a distinct clade within the family *Halobacteriaceae* suggests a relatively recent origin of this complex phenotype. It remains to be shown whether complex cellular differentiation, which evolved independently in several groups of bacteria, also independently emerged in other archaea.

### Taxonomic description
As mentioned above, phylogenetic analysis of 16S rRNA and concatenated conserved genes indicated that strain YIM 93972 formed a clade with the closest related species *Halocatena pleomorpha* within family *Halobacteriaceae*. The average nucleotide identity based on BLAST (ANIb) value between YIM 93972 and *H. pleomorpha*

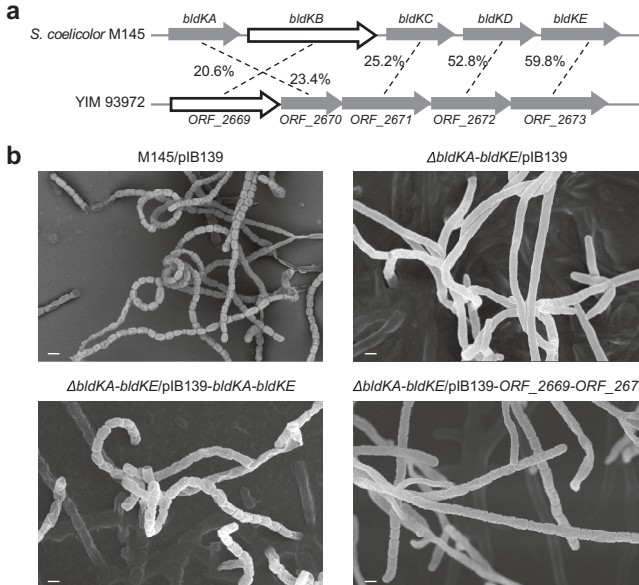

**Fig. 6 | Role of peptide permeases in morphological differentiation: complementation of a *bldKA-KE* deletion mutant of *Streptomyces coelicolor* with the homologous operon of YIM 93972. a** Comparison of the *bldK* operon of *S. coelicolor* M145 with *ORF_2669-ORF_2673* of YIM 93972; arrows indicate the size and directions of genes; percentage of amino acid sequence identity is indicated for each gene. **b** Comparison of mycelia phenotypes among M145/pIB139, *ΔbldKA-KE*/pIB139, *ΔbldKA-KE*/pIB139-*bldKA-KE*, and *ΔbldKA-KE*/pIB139-*ORF_2669-ORF_2673* using scanning EM (bar, 5 μm). Deletion of *bldKA-KE* in M145 abrogated the development of aerial mycelia and spores; this phenotype was restored to near wild-type levels by complementation with *bldKA-KE* or *ORF_2669-ORF_2673* (*n* = 3).

SPP-AMP-1[T] was 70.59%, which was far below the cut-off values <75 % for genus demarcation in the class *Halobacteria*[42]. Further, the low AAI (68.4%) between YIM 93972 and *H. pleomorpha* SPP-AMP-1[T] confirms the novelty of YIM 93972 at the genus level[42]. The substrate mycelium of YIM 93972 formed sporangia, and the aerial mycelium formed spore chains at maturity and the spores were cylindrical with wrinkled surfaces. The respiratory quinones of strain YIM 93972 are MK-8 and MK-8(H₂) (Supplementary Fig. 14). By virtue of having a different isomer for MK-8 (H₂), the respiratory quinones of YIM 93972 differ from those of *H. pleomorpha* SPP-AMP-1[T], which contains only menaquinone MK-8. The major polar lipids of the strain were chromatographically identified as phosphatidylglycerol, phosphatidylglycerolphosphate methyl ester and five unidentified glycolipids, while *H. pleomorpha* SPP-AMP-1[T] has phosphatidylglycerol, phosphatidylglycerolphosphate methyl ester, glycosyl mannosyl glucosyl diether and sulphated glycosyl mannosyl glucosyl diether. Besides the phylogenetic and morphological differences, the isolate is differentiated from the closely related genera by its chemotaxonomic markers (Supplementary Data 1). Based on the above results, YIM 93972[T] merits representation as a new species in a new genus within the family *Halobacteriaceae*, for which the name *Actinoarchaeum halophilum* gen. nov., sp. nov. is herewith proposed.

### Description of *Actinoarchaeum* gen. nov

*Actinoarchaeum* (Ac.tino.ar.chae'um. Gr. n. *aktis -inos*, a ray; N.L. neut. n. *archaeum* (from Gr. adj. *archaios -ê -on*, ancient), archaeon; N.L. neut. n. *Actinoarchaeum* ray archaeon, referring to the radial arrangement of filaments).

Aerobic, extremely halophilic, *Streptomyces*-like colonies that produce brown substrate mycelia forms terminal sporangia with white aerial mycelia and white spore on modified ISP 4 medium. The major polar lipids are phosphatidylglycerol, phosphatidylglycerol phosphate

methyl ester and five unidentified glycolipids. The predominant menaquinones are MK-8 and MK-8(H₂). The G + C content of the genomic DNA is about 56.3 mol%. The type species is *Actinoarchaeum halophilum*. Recommended three-letter abbreviation: *Aah*.

### Description of *Actinoarchaeum halophilum* sp. nov

*Actinoarchaeum halophilum* (ha.lo'phi.lum. Gr. n. *hals* halos, salt; N.L. adj. *philus -a -um*, from Gr. adj. *philos -ê* -on, friend, loving; N.L. neut. adj. *halophilum*, salt-loving).

Morphological, chemotaxonomic, and general characteristics are as given above for the genus. Cells require 2.1–6.0 M NaCl, pH 6.0–9.0, 25–50 °C and Mg²⁺ (0.01–0.7 M) for growth. Optimal growth occurs at 3.8–4.2 M NaCl, pH 7.0–7.5, 40–45 °C. Oxidase-weakly positive and catalase-negative. Positive for nitrate reduction, hydrolysis of Tweens (20, 40, 60, 80), casein and gelatin; while negative for productions of indole and H₂S. Acetate, citrate, dextrin, fructose, fumarate, glucose, glycerol, malate, mannitol, mannose, pyruvate, rhamnose, succinate, sucrose and trehalose are utilized as sole carbon sources, but galactose, lactate, lactose, xylitol and xylose are not. Acid is not produced from any of sole carbon sources stated above. The strain contains MK-8 and MK-8(H₂). Polar lipids include PG, PGP-Me and 5GLs. The genomic DNA G+C content is 56.3%. The type strain is YIM 93972[T] (=DSM 46868[T] = CGMCC 1.17467[T]), isolated from a soil sample from Salt Lake in Xinjiang Uygur Autonomous Region of China.

## Methods

### Isolation, identification, morphological observation

The sediment samples were collected from Qijiaojing Salt Lake in Xinjiang Uygur Autonomous Region, China. Isolation was done using the standard dilution-plating technique on Modified Gause (MG) medium[11] containing (g/L): soluble starch, 5; lotus root starch, 5; KNO₃, 1; MgSO₄·7H₂O, 0.5; K₂HPO₄, 0.5; NaCl, 200; agar, 20; 1 mL trace solution (2% FeSO₄·7H₂O; 1% MnCl₂·4H₂O; 1% ZnSO₄·7H₂O; 1% CuSO₄·5H₂O; pH adjusted to 7.2). Plates were incubated at 37 °C for at least 6 weeks. All isolates with actinobacteria-like filamentous colonies were collected. In the identification of these isolates based on 16S rRNA gene, one isolate designated YIM 93972 was paid more attention, because its 16S rRNA gene could not be amplified using universal primers for the bacterial 16S rRNA gene, but the primers for archaeal 16S rRNA gene worked. Cell morphology of strain YIM 93972 was further examined using a scanning electron microscopy (Quanta 2000, FEI, Hillsboro, OR, USA) at different concentrations of NaCl. Cells from a 20-day-old culture of strain YIM 93972 were fixed for 30 min in OsO₄ vapour on ice, and filtered onto 2 μm isopore membrane filters (Millipore, Tokyo, Japan) modified by the method of Schubert et al[43]. The fixed cultures were dehydrated in a series of 50–100% ethanol mixtures, critical-point-dried in CO₂ and sputter-coated by 10 nm of gold-palladium. Cells were imaged using a XL30 ESEM-TMP (Philips-FEI, Eindhoven, Holland) at 3 kV. For testing the temperature tolerance of spores, spores were separated from substrate hyphae by virtue of a nitrocellulose transfer membrane (see below for details). Spore and mechanically broken substrate hyphae were used for preparing suspension (2 OD) and bathed at 60, 70, 80 and 90 °C for 15 min (control, 37 °C). Then, 100 μL suspension was spread on solid ISP 4 medium with 20% NaCl, and cultured for 30 days. Images were collected every 10 days.

For obtaining more strains like YIM 93972, the same isolation was performed on the soil samples collected from the Aiding, Large south, Dabancheng East, and Uzun Brac Salt Lakes in Xinjiang Uygur Autonomous region. Finally, five haloarchaea strains were isolated and confirmed with morphological differentiation (cultured on MG plates containing 20% NaCl at 37 °C for 3-4 week). Among them, YIM A00010, YIM A00011, and YIM A00014 were isolated from the Aiding Salt Lake samples; YIM A00012 and YIM A00013 were isolated from the Uzun Brac Salt Lake samples. For further understanding the diversity of YIM

93972 and its relatives in these four salt lakes, the 16S rRNA genes were amplified by using a pair of YIM 93972-specific primers (Forward primer: 5′ -GGGCGTCCAGCGGAAACC-3′ ; Reverse primer: 5′ -CCAT-CAGCCTGACTGTCAT-3′), directly from the total DNA of 15 soil samples. The PCR products were isolated from 2% agarose gel and purified using the AxyPrep DNA Gel Extraction Kit (Axygen Biosciences, Union City, CA, USA) according to manufacturer's instructions and quantified using Quantus™ Fluorometer (Promega, Madison, WA, USA). Purified amplicons were pooled in equimolar amounts and paired-end sequenced on an Illumina MiSeq PE300 platform (Illumina, San Diego, CA, USA) according to the standard protocols by Majorbio Bio-Pharm Technology Co. Ltd. (Shanghai, China)[44].

### Genomic sequencing

The total genomic DNA from strain YIM 93972 was extracted using the DNeasy Power Soil Kit (QIAGEN, Hilden, Germany). The quality of DNA was assessed by 1% agarose gel electrophoresis, and the purity were measured using NanoDrop™ 2000 spectrophotometer (ThermoFisher Scientific, Waltham, MA, USA). Qualified genomic DNA (5 μg) was subjected to SMRT sequencing using a Pacific Biosciences RSII sequencer (Pacific Biosciences, Menlo Park, CA, USA). The Hierarchical Genome Assembly Process (HGAP, v2.3.0) pipeline was used to a generate high quality de novo assembly of the genome with default parameters[45]. The genomic DNAs of other five isolates were extracted as well as, and the quality was assessed using the same procedure. Then, the genomic DNAs were split into two fetches and sequenced using Oxford Nanopore (PromethION). The raw datasets were filtered, the subreads were assembled by Canu (v1.5)[46], and further corrected by Pilon (v1.22)[47]. For further calibration the errors of the assembled genome using second generation sequencing datasets by Illumina NovaSeq 6000, and high-quality reads were obtained to generate one or more contigs without gaps. The rRNAs and tRNAs of YIM 93972 were predicted using RNAmmer (v1.2)[48] and tRNAscan-SE (v1.3.1)[49], respectively. The gene predication and annotation were performed using Prokka (v1.14.6). The circular maps were generated by DNA-Plotter (v1.11)[50]. The detail assembled and annotated results are shown in Supplementary Data 2a-f.

### Comparative genome and phylogenome

The complete genomes of 122 *Halobacteria* isolates were downloaded from NCBI Genomes site. Additionally, contig level genome assemblies of a pleomorphic haloarchaeon *Halocatena pleomorpha* SPP-AMP-1$^T$ (GCF_003862495.1) and a closely related non-morphogenetic haloarchaeon *Halomarina oriensis* JCM 16495$^T$ (GCF_009791395.1), were downloaded from the same source. Protein sequences, encoded by these 124 publicly available *Halobacteria* genomes, as well as those encoded in the six genomes, obtained in this study, were analyzed to establish the orthology relationships between their genes (Supplementary_data_file_1). Initially, all protein sequences were clustered using MMseqs2 (v14-7e284)[51] with sequence similarity threshold of 0.5; then the clusters were further refined via several iterations of the following procedure:

- cluster alignments, obtained using MUSCLE (v5)[52] were compared to each other using HHSEARCH (v3.3.0);[53] clusters, displaying full-length similarity were merged together;
- approximate ML phylogenetic trees were reconstructed using FastTree (v 2.1.11)[54] for the merged cluster alignments[52] and rooted by mid-point; trees were parsed into subtrees, maximizing the ratio of taxonomic coverage (number of the distinct genome assemblies in the subtree) to the paralogy index (average number of sequences per assembly);

This procedure produced a set of 14,870 clusters (excluding singleton sequences) of orthologous genes (haloCOGs) that subsequently served for comparative evolutionary genomics analysis of

*Halobacteria* genes. Functional annotation of haloCOGs was obtained by comparing haloCOG alignments to CDD[55] and arCOG[56] sequence profiles using HHSEARCH[53].

268 haloCOGs with full complement of 130 genomes and at most 4 additional paralogs were used to determine the genome-level phylogeny of *Halobacteria* (Supplementary Data 3, Supplementary_data_file_2). When paralogs were present, the index ortholog was selected based on the BLOSUM62[57] alignment score between the paralogs and the alignment consensus. Columns in the alignments of these 268 haloCOGs were filtered[58] for maximum fraction of gaps (0.667) and minimum homogeneity (0.05). The concatenated alignment contained 71,586 amino-acid sites. Phylogenetic tree was reconstructed using IQ-Tree (v2.2.0)[59] under the LG+F+R10 model, selected by the built-in model finder and rooted according to Rinke et al. 2021 (Supplementary _data_file_2)[60].

The history of gene gains and losses (Supplementary Data 4b, c) in *Halobacteria* was reconstructed from the haloCOG phyletic patterns and the *Halobacteria* phylogenetic tree using GLOOME (v201305)[61]. Gains or losses of a gene on a particular edge of a phylogenetic tree were inferred from changes in the posterior probability of this gene presence between the ancestral and the descendant genomes at the respective ends of this edge; a change in probability exceeding 0.5 in magnitude was interpreted as a likely gain or loss of the gene (Supplementary Data 4 and 4c).

### Morphologically defective mutants screening and genome resequencing

To obtain the poorly differentiated and undifferentiated mutants, a nitroso-guanidine (NTG) based mutagenesis experiment was applied to the wild type of YIM 93972. Spore suspensions were prepared by adding sterile glass beads (5 mm diameter; 1 g per 10 mL medium) in ISP 4 medium containing (g/L): (soluble starch, 10; K$_2$PO$_4$, 1; MgSO$_4$, 1; (NH$_4$)$_2$SO$_4$, 2; NaCl, 250) at 37 °C for 3 days, and then adjusted to 10$^8$ spores/mL. NTG treatment was carried out by incubating spore suspension with varied concentrations of NTG from 0.2 to 1 mg/mL at 30 °C for 30 min on an incubator shook at 90 g. Samples of 10 mL were harvested at 2400 g for 5 min, and spores were washed three times with 20 mL of 20% sterile NaCl. The spore lethality rates were measured by the LIVE/DEAD™ *Bac*Light™ bacterial viability kit (Thermo-Fisher Scientific, Waltham, MA, USA) as described previously[62]. The NTG treated spores with ~85% lethality rates were diluted to 10$^2$ and 10$^3$ per milliliter, and then 100 μL aliquots were equally spread on ISP 4 medium plates containing 25% NaCl and then incubated at 37 °C for 28 days. The mutants poorly differentiated (transitional) and that undifferentiated (bald) were cultured and passaged four times for further multi-omics study (Supplementary Fig. 10a). The morphological phenotype of many mutants recovered in the next sub-cultivation. So, after 4 generations, only 5 transitional mutants and 3 bald mutants remained with stable morphological mutant phenotypes. To further confirm the morphological mutation of strain YIM 93972, we also checked the cell morphology by scanning electron microscopy as described above. Compared with wild type colonies, two transitional colonies had branched substrate mycelia and very rare aerial mycelia. However, the three bald colonies only had branched substrate mycelia (Supplementary Fig. 10b).

### Comparative transcriptome

For collecting the aerial and substrate hyphae of wild type, viable spores 10$^7$ per mL of strain YIM 93972 were collected after full sporulation of the aerial hyphae (28 days of solid medium incubation) and spread on a layer of 0.20 μm BioTrace NT Nitrocellulose Transfer Membrane (Pall China, Beijing, China) covered on a medium ISP 4 containing 20% NaCl and 2% agar in 85 mm diameter Petri dish. The plates were cultured at 37 °C for 21 days. Totally, 96 culture plates were prepared, which were split into 6 groups as biological replicates.

The aerial hyphae grown on nitrocellulose membrane, and the substrate hyphae penetrated through nitrocellulose membrane were recovered by scraping with a plain spatula, respectively. For two transitional and three bald mutants, the equal amounts of mechanically broken hyphae were spread on the same plates as described above. Totally 192 culture plates were prepared for each mutant, which were split into 6 groups as biological replicates. The substrate hyphae were recovered at 28 days. Totally, 6 pooled hyphae were collected for each strain, half of them were used for transcriptomic analysis, and another half were used for proteomic analysis.

The total RNA was extracted and analyzed as described previously[63]. Briefly, total RNA was extracted with Trizol method. The TruSeq™ Stranded Total RNA Library Prep Kit (Illumina, San Diego, CA, USA) was used to prepare the cDNA libraries, followed by paired-end 100 bp sequencing on an Illumina HiSeq 2000 system (Illumina, San Diego, CA, USA). The RNA-Seq reads per sample were mapped to the reference using Bowti2 (v 2.4.2)[64]. RSEM[65] (v1.3.3) was used to estimate TPM. SNP analysis was performed by SAM tools[66] (v1.12). The detailed quality control was showed in Supplementary Data 6 and Fig. 3b. Genes with ratios greater than 1.5-fold and $p$-value smaller than 0.05 (Student's $t$-test) were considered as regulated and used for further bioinformatics analysis (Fig. 3d and Supplementary Data 9).

## Comparative proteome

For proteomic analysis, the aerial hyphae and substrate hyphae were suspended in lysis buffer [9 M Urea, 10 mM Tris-HCl (pH 8.0), 30 mM NaCl, 5 mM iodoacetamide (IAA), 5 mM $Na_4P_2O_7$, 100 mM $Na_2HPO_4$ (pH 8.0), 1 mM NaF, 1 mM $Na_3VO_4$, 1 mM sodium glycerophosphate, 1% phosphatase inhibitor cocktail 2, 1% phosphatase inhibitor cocktail 3, EDTA-free protease inhibitor cocktail (1 tablet/10 mL lysis buffer)] and disrupted by a Soniprep sonicator (Scientz, Ningbo, Zhejiang, China) for 10 min (2 s on and 4 s off, amplitude 30%) as described[67]. The lysates were centrifuged at 16,200 $g$ for 10 min at 4 °C to remove debris. The quality and concentration of extracted total cell lysates (TCL) were detected by a 10% SDS-PAGE (Supplementary Fig. 11a). The technical replicates were designed in wild, transitional, and bald groups, respectively.

Same amount of pooled proteins (120 µg) from each sample was reduced with 5 mM of dithiothreitol (DTT) at 45 °C for 30 min, followed by alkylation with 10 mM of iodoacetamide (IAA) at room temperature for 30 min. The alkylated samples were precleaned with a 10% SDS-PAGE (0.7 cm)[68], and in-gel digested with 12.5 ng/µL trypsin[69] at 37 °C for 14 h. Equal amount of peptides from each differentiated condition were used for 10-plex TMT labeling (ThermoFisher Scientific, Waltham, MA, USA). TMT labeling was carried out as follows: wild substrate hyphae (W-SH) technical replicates were labeled with 126 and 127[N]; wild type aerial hyphae (W-AH) with 127[C]; transitional substrate hyphae (T-SH) technical replicates with 128[N] and 128[C], T-SH biological replicate with 129[N]; bald substrate hyphae (B-SH) technical replicates with 130[N] and 130[C], B-SH biological replicates with 129[C] and 131 following the manufacturer's protocol (Fig. 4a). The labeled peptides were mixed and dried with a vacuum dryer (LABCONCO CentriVap, Kansas City, MO, USA).

The mixed TMT labeled peptides were fractionated by a Durashell $C_{18}$ high pH reverse phase (RP) column (150 Å, 5 µm, 4.6 × 250 mm², Bonna-Agela Technologies Inc., Newark, DE, USA) on a Rigol L-3120 HPLC system (Beijing, China) as described previously[67]. Briefly, the solvent gradients comprised buffer A (98% double distilled $H_2O$ and 2% ACN, pH 10, adjusted by ammonium hydroxide) and buffer B (2% double distilled $H_2O$ and 98% ACN, pH 10). The mixed samples were dissolved in buffer A. After loading, the peptides were separated with a 60 min linear gradient (0% B for 5 min, 0-3% B for 3 min, 3–22% B for 37 min, 22–32% B for 10 min, 32-90% B for 1 min, 90% B for 2 min, and 100% B for 2 min). The LC flow rate was set at 0.7 mL/min. The column was maintained at 45 °C. Eluent was collected every 1 min

(Supplementary Fig. 11b). The 60 fractions were dried and concatenated to 10 fractions (Supplementary Fig. 11c). The combined peptides were subjected to LC−MS/MS analysis.

The fractions were analyzed on a Q Exactive HF mass spectrometer (ThermoFisher Scientific, Waltham, MA, USA) after Easy-Nano LC 1200 separation (ThermoFisher Scientific, Waltham, MA, USA). Briefly, samples were loaded onto a self-packed capillary column (75 µm i.d. × 50 cm, 1.9 µm $C_{18}$) and eluted with a 135 min linear gradient (4–8% B for 13 min, 8–25% B for 86 min, 20–50% B for 21 min, 50–90% B for 3 min, 90% B for 12 min). Full MS scans were performed with $m/z$ ranges of 375–1,400 at a resolution of $1.2 \times 10^5$; the maximum injection time (MIT) was 80 ms, and the automatic gain control (AGC) was set to $3.0 \times 10^6$. For the MS/MS scans, the 15 most intense peptide ions with charge states of 2 to 6 were subjected to fragmentation via higher energy collision-induced dissociation (HCD) (AGC: $1 \times 10^5$, MIT: 100 ms, Resolution: $6 \times 10^4$). Dynamic exclusion was set as 30 s.

All of the raw files were searched with MaxQuant (v1.5.6.0) against the protein database from strain YIM 93972 (3744 entries) along with 245 common contaminant protein sequences (http://www.maxquant.org). Fully tryptic peptides with up to two miss cleavage sites were allowed. Oxidation of methionine was set as dynamic modification, whereas carbamidomethylation of cysteine and TMT modification at peptide N-terminus and lysine were set as static modifications. Figure 4b showed the MS identification and quantification. The detail quality control was showed in Supplementary Fig. 11d, e. Nine protein expression clusters were showed in Fig. 4e. Protein changes greater than 1.5-fold and $p$-value smaller than 0.05 (Significance A[70]) were considered as regulated, which were further used for bioinformatics analysis (Fig. 4e and Supplementary Data 12).

## Heterologous expression and functional analysis

To characterize the putative oligopeptide transporter encoded by *ORF_2669-ORF_2673* from YIM 93972, a bialaphos (from 0.1 µg/mL to 1.0 µg/mL with 0.1 as interval) resistance test[34, 67] was performed using solid ISP 4 medium. Then, the heterologous expression of genes *ORF_2669-ORF_2673* in *Streptomyces coelicolor* M145[71] with a deletion of gene cluster *bldKA-KE* was conducted. The Δ*bldKA-KE* mutant was constructed by the CRISPR-Cas9 genome editing method as previously described[72]. The strains and plasmids used were listed in Supplementary Data 15, and the primers in Supplementary Data 16. Briefly, the upstream (1139 bp) and downstream regions (1310 bp) of *bldKA-KE* were amplified using the primer pairs *bldKA-KE*-up-F/R and *bldKA-KE* -down-F/R, respectively. The single guide RNA (sgRNA) transcription cassette was obtained from the plasmid pKCCas9dO with the primers *bldKA-KE*-gRNA F/R. Then, three above fragments were assembled by an overlapping PCR with the primers *bldKA-KE*-gRNA-F and *bldKA-KE*-down-R, followed by double digestion with *Spe* I and *Hind* III. The assembled fragment was cloned into pKCCas9dO and was introduced into the wild strain M145 by conjugal transfer. The constructed strain was subsequently grown on solid MS medium without apramycin at 37 °C to remove the plasmid pKCcas9-*bldKA-KE*. The correct double crossover colonies were verified with the primers *bldKA-KE*-J-F/R, yielding the Δ*bldKA-KE* mutant strain.

Using M145 genomic DNA as the template, the *bldKA-KE* from M145 and *ORF_2669-ORF_2673* from YIM 93972 were obtained with the primer pairs *bldKA-KE*-F/R and *ORF_2669-ORF_2673* respectively. The amplified products were cloned in the pIB139 vector to generate the complemented plasmids pIB139-*bldKA-KE* and pIB139-*ORF_2669-ORF_2673*. The obtained plasmids were introduced into the Δ*bldKA-KE* mutant, resulting in two complemented strains, the Δ*bldKA-KE*/pIB139-*bldKA-KE* and Δ*bldKA-KE*/pIB139-*ORF_2669-ORF_2673* strains. The empty vector pIB139 was transferred into the M145 strain, generating the positive control. To evaluate the role of oligopeptide ABC

transporter *ORF_2669-ORF_2673* for controlling mycelium differentiation as *bldKA-KE* in *S. coelicolor*, the same viable spores of the four strains, M145/pIB139, Δ*bldKA-KE*/pIB139, Δ*bldKA-KE*/pIB139-*bldKA-KE* and Δ*bldKA-KE*/pIB139-*ORF_2669-ORF_2673* were inoculated and grown on an ISP 4 agar plate at 37 °C. About 700 visible clones were grown on each agar plate. The substrate and aerial mycelium from a 78 h old culture of the above four constructed strains were examined by a scanning electron microscopy (Nova NanoSEM 450, FEI, USA).

### Characterization of cell wall S-layer proteins of YIM 93972

The S-layer proteins of YIM 93972 were extracted as previously described[73]. The mid-log phase cells were harvested by centrifugation at 3500 *g*, and pellets were resuspended in cell lysis buffer containing 20 mM Tris-HCl (pH 6.5), 1.0 mM phenylmethylsulfonyl fluoride (PMSF), and 1x protease inhibitor cocktail (Roche, Basel, Switzerland), and disrupted by a Soniprep sonicator (2-s on and 4-s off, amplitude 25%) for 10 min. Cell debris were removed by centrifugation at 4000 *g* for 15 min at 4 °C. The supernatant was further ultracentrifuged for 1 h at 250,000 *g* and 4 °C. Then the pellet was dissolved in a buffer containing 4% SDS, 50 mM Tris (pH 8.0) and 20 mM DTT to obtain cell wall proteins. 20 µg of cell wall proteins were reduced with 5 mM of DTT at 45 °C for 30 min and alkylated with 10 mM of IAA at room temperature for 30 min. After separated with a 12% SDS-PAGE for 5 cm, the whole lane was sliced into 25 fractions based on the distinguishing bands on the Coomassie blue G-250 stained gel. The gel pieces were further excised into 1 cubic millimeter gels and digested in-gel with trypsin (12 ng/µL) at 37 °C for 14 h. The tryptic peptides were analyzed by an LTQ-Orbitrap Velos mass spectrometer after high pH RP chromatography separation. The acquired raw files were then submitted to the MaxQuant (v1.5.6.0) against the protein database from strain YIM 93972 (3744 entries) along with common contaminant protein sequences. Two S-layer proteins were detected by peptide profile matching with the annotated protein database of YIM 93972 based on peptide numbers ≥2, including ORF_1400 and ORF_1704 (Supplementary Fig. 9a–d).

### Statistics and reproducibility

The strain culture, scanning EM image, transmission EM image, were performed at least three independent experiments. Statistical differences between two groups in transcriptomic and proteomic datasets were analyzed using two-tailed unpaired *t*-tests (inter-group) and Significance A[70] (intra-group), respectively. *P* < 0.05 was considered statistically significant. Data statical analysis and visualization were realized by R (v4.1.2).

### Reporting summary

Further information on research design is available in the Nature Portfolio Reporting Summary linked to this article.

## Data availability

The genome and transcriptome data generated in this study have been deposited in GenBank under accession codes CP071306-CP071310 and GSE168067, respectively. The raw reads of amplicon sequencing are deposited in GenBank, under accession number PRJNA703326. All proteomic MS data sets can be obtained from iProX[74] database with identifier IPX0002724000, or ProteomeXchange with identifier PXD023481. Raw reads of genome for strains YIM_A00010, YIM_A00011, YIM_A00012, YIM_A00013, and YIM_A00014 were deposited in GenBank under accession codes JAKCFJ000000000, JAKCFK000000000, JAKCFL000000000, JAKCFM000000000, and JAKCFN000000000, respectively. The complete Halobacterial COG data archive (supplementary_data_file_1.tgz) and the phylogenetic trees and alignments archive (supplementary_data_file_2.tgz) can be downloaded from https://ftp.ncbi.nih.gov/pub/wolf/_suppl/halo22/.

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

## Acknowledgements

We are indebted to Dr. Wen-Jun Li for support in the early stage of this project. The authors thank Dr. Xiu-Zhu Dong, Yong-Jun Liu, Lei Song and Man Cai for strains, reagents, and discussion. We thank Dr. Lingyan Li for discussion of S-layer protein extraction. We thank Professor Aharon Oren for the etymology of new generic name and of new epithet, and Professors Leonard Krall, Felix Cheung and Xu-Na Wu for critical reading and editing. We also thank Shu-Jia Wu, Jin-Shuai Sun, Wen-Hui Wu, Jie Ma and Xue Wang for experiments and data processing. P.X., G.P.Z., S.K.T., X.Y.Z., Y.Z., B.B.L. and F.C.H. are supported by the Chinese National Basic Research Programs (2022YFA1304600 and 2020YFE0202200), the Innovation Foundation of Medicine (2022F12010, 20SWAQX34 and AWS17J008), the National Natural Science Foundation of China (32141003, 31901037, 31870824, 91839302, 32071431, 31760003, 32060003, 32070668, 92151001 and 31800001), the Beijing-Tianjin-Hebei Basic Research Cooperation Project (J200001), CAMS Innovation Fund for Medical Sciences (2019-I2M-5-017, 2019-12M-5-063 and 2022-I2M-C&T-B-082), the Foundation of State Key Lab of Proteomics (SKLP-C202002, 2021-NCPSB-001, SKLP-K201704, & SKLP-K201901) and Major Science and Technology Projects of Yunnan Province (202002AA100007). K.S.M., Y.I.W. and E.V.K. are supported by intramural funds of the US Department of Health and Human Services (National Institutes of Health, National Library of Medicine).

## Author contributions

P.X., S.K.T., E.V.K and G.P.Z. designed experiments. K.L. and Y.W. collected the environmental samples. S.K.T. and B.B.L. performed the strain isolation, polyphasic taxonomic analysis, culture of YIM 93972 and all the other 6 morphological differentiation haloarchaeon strains, and SSU rRNA verification with the help of Y.R.Z., E.Y.L., Y.Z.F., M.X.X., Z.Q.L., X.Z., R.L., M.Y., L.L.Y., T.W.G., H.L.C., Z.K.Z. and T.S.T. M.X.X. and Z.Q.L. performed heat stress experiment under the direction of T.S.T., P.X., Y.Z. and S.K.T. X.M.C performed the mutagenesis experiments of YIM 93972. H.J.Z. performed the genomic sequencing of the wild and mutated strains. X.Y.Z., K.S.M. and Y.I.W. performed the genome annotation and phylogenomic analysis. Y.Z. performed all of the proteomic experiments and data analysis under the help of S.H.J., T.Z., P.R.C., J.H.S., C.C., L.C. and H.Y.G. X.Y.Z. and Y.Z. performed transcriptomic analysis, and arranged all figures and tables. P.X., Z.P.Z., Y.Z. and K.S.M. performed transcriptional regulator analysis. Z.P.Z. and S.H.J. helped data analysis and uploading. Y.H.L. and G.S.Z. constructed all the recombinant plasmids and gene deletion strains for gene complementation experiments. F.C.H. helped article frame arrangement about six cellular differentiational halophilic archaea. P.X., Y.Z., G.S.Z., Y.H.L. and S.K.T. designed and performed spore-genetic gene cluster verification with the help of G.P.Z., E.V.K. and K.S.M. X.Y.Z., Y.Z., K.S.M., P.X., E.V.K. and G.P.Z. wrote the manuscript with the help of all authors.

## Competing interests

The authors declare no competing interests.

## Additional information

[1]Yunnan Institute of Microbiology, Key Laboratory for Microbial Resources of the Ministry of Education, School of Life Sciences, Yunnan University, Kunming 650091, China. [2]State Key Laboratory of Proteomics, Beijing Proteome Research Center, National Center for Protein Sciences Beijing, Research Unit of Proteomics & Research and Development of New Drug , Research Unit of Proteomics Driven Cancer Precision Medicine, Chinese Academy of Medical Sciences, Institute of Lifeomics, Beijing 102206, China. [3]National Center for Biotechnology Information, National Library of Medicine, National Institutes of Health, 8600 Rockville Pike, Bethesda, MD 20894, USA. [4]Henan Key Laboratory of Industrial Microbial Resources and Fermentation Technology, College of Biological and Chemical Engineering, Nanyang Institute of Technology, Nanyang 473004, China. [5]College of Life Sciences, Shanghai Normal University, Shanghai 200234, China. [6]Shanghai-MOST Key Laboratory of Health and Disease Genomics, Chinese National Human Genome Center at Shanghai and Shanghai Institute for Biomedical and Pharmaceutical Technologies, Shanghai 201203, China. [7]Hebei Province Key Lab of Research and Application on Microbial Diversity, College of Life Sciences, Hebei University, Hebei 071002, China. [8]Key Laboratory of Desert and Desertification, Northwest Institute of Eco-Environment and Resources, Chinese Academy of Sciences, Lanzhou 730000, China. [9]Xinjiang Institute of Microbiology, Xinjiang Academy of Agricultural Science, Urumqi 830091, China. [10]Biotechnology and Genetic Germplasm Resources Research Institute, Yunnan Academy of Agricultural Sciences, Kunming 650205, China. [11]Department of Microbiology, Key Laboratory of Combinatorial Biosynthesis and Drug Discovery of Ministry of Education, School of Pharmaceutical Sciences, Wuhan University, Wuhan 430072, China. [12]College of Food and Biological Engineering, Xihua University, Chengdu 610039, China. [13]School of Food and Biological Engineering, Jiangsu University, Zhenjiang 212013, China. [14]Key Laboratory of Medical Molecular Virology (MOE/NHC/CAMS), School of Basic Medical Sciences and Department of Microbiology and Microbial Engineering, School of Life Sciences, Fudan University, Shanghai 200032, China. [15]Guizhou University, School of Medicine, Guiyang 550025, China. [16]State Key Laboratory of Dampness Syndrome of Chinese Medicine, The Second Affiliated Hospital of Guangzhou University of Chinese Medicine, Guangzhou 510120, China. [17]These authors contributed equally: Shu-Kun Tang, Xiao-Yang Zhi, Yao Zhang, Kira S. Makarova, Bing-Bing Liu. ✉e-mail: tangshukun@ynu.edu.cn; koonin@ncbi.nlm.nih.gov; gpzhao@sibs.ac.cn; xuping@ncpsb.org.cn

