## [Peer Review File · Nature Communications]

Cellular differentiation into hyphae and spores in halophilic archaeaReviewer #1 (Remarks to the Author):

The authors report their isolation and identification of a halophilic archaeon exhibiting extensive morphological differentiation, including the presence of hyphae and spores. Multiple signature gene losses and gains are reported which may support the mycelial lifestyle of this organism. Transcriptomics and proteomics revealed a CDC48 ATPase (ORF_0238) may be involved in this morphological differentiation. Most interestingly, non-differentiating mutants obtained through chemical mutagenesis strongly implicated an oligopeptide transporter in areal hyphae differentiation. Specific comments:

Line 56: Complex morphological differentiation and pleomorphis has been well described in archaea previously (and quite complex, certainly at a molecular level), e.g. swimming or biofilm-related cell morphology transformations in haloarchaea, but what the authors might mean here is a metamorphosis/transformational type of differentiation, where the cell types are obviously and radically different in many different ways. This terminology should be corrected and could be made clearer and consistent throughout the paper. The authors should not use the term pleomorphology to describe their different cell types, because what they are claiming is a much more substantial state of cellular differentiation than how 'pleomorphology' is widely used in microbiology. (refer to textbook usages). Otherwise, the paper may attract additional criticism for claiming novelty that is not quite true.

In this regard, the authors could consider revising the bold claim of the paper's title.

Line 59: 'unlike other archaea' – the authors should at least qualify 'known archaea'.

Line 136: The authors state on that when *H. pleomorpha* was plated onto ISP4 medium supplemented with NaCl, that similar morphological differentiation was observed to the newly isolated strains and that this may be common across Halobacteria (contradicting Line 59 comment). Is it possible that the chosen medium can induce morphological differentiation in many Halobacteria – majority of which are already described as pleomorphic (e.g. the model *H. salinarum*, *H. volcanii* etc)? Can the authors show evidence that such differentiation is limited to the phylogenetic branch implicated, as assumed?

How likely is it that these Haloarchaea (including YIM 93972) undergo morphological differentiation in their ecological environments and what benefits might this provide them with? This may be worth discussing briefly, or referring to environmental samples if possible, to show better evidence of the relevance of the different cell types.

Are the apparent spores shown to be dormant and non-replicative? Furthermore, do they show substantially elevated broad-spectrum stress resistance, as expected for the definition of a spore?

Line 150: *Natronomonas* is not shown in Figure 2b as part of the sister clade of the pleomorphic clade but is referenced in the text.

Line 163: As noted, Cetz proteins are necessary for controlling pleomorphic shape changes in other Halobacteria. In addition, supplementary table 4C shows the loss of arCOG06187 corresponding to a possible MreB (or actin-like) homologue. Why would the loss of these cell shape controlling proteins be associated with or promote the pleomorphic phenotype observed in the current study? Are any other cytoskeletal homologues present that may contribute to pleomorphism in these isolates?

Line 165: The type IV pili system typified by the PilB3 locus of *H. volcanii* is present outside of Halobacteria as well, including Methanomicrobia and Archaeoglobi according to Makarova et al., 2016 "Diversity and Evolution of Type IV pili Systems in Archaea", supplementary Figure 3.

Line 169: Are there any examples in the literature that CDC48 family ATPases contribute specifically to pleomorphism or shape changes including hyphae formation or sporulation in other organisms?

Line 213: The authors defined the bald mutants as having no aerial hyphae but do not suggest or

show that these mutants are also sporulation deficient. I agree that ORF_0238 could potentially be important for morphological differentiation generally, but its role may be direct or indirect and having knock-on affects on morphological differentiation. The conclusion that ORF_0238 itself is functionally important for morphological differentiation of hyphae and sporulation may be too strong given the evidence – especially if there are no specific examples of CDC48 family ATPases contributing to pleomorphism, morphological differentiation, or sporulation in the literature. If there are homologues of ORF_0238 in model bacteria or archaea it may be informative to conduct complementation studies similar to those in Figure 5 to strengthen the evidence.

Line 247 and 497: Complementation of the *S. coelicolor* M145 oligopeptide permease deletion with both YIM 93972 ORF_2669-2673 and the endogenous *S. coelicolor* oligopeptide permease does appear to restore mycelia and spores, however the morphology of the spores in the complementation strains look potentially larger and less round or differentiated compared to M145/pIB139, perhaps indicating only partial complementation rather than full restoration of the sporulation defect.

Reviewer #2 (Remarks to the Author):

The fascinating study of Tang et al. report a novel mode of development for a haloarchaeal species that physically resembles the classical sporulating life cycle of *Streptomyces* bacteria. The sporulating properties of this organism seem confined to a specific clade within the Halobacteria, and seem to be associated with genes unique to these bacteria. The absence of genetic tools for this organism precluded any in-depth genetic analyses, but mutational studies successfully yielded mutants defective in their ability to raise aerial hyphae. The authors followed up these mutational analyses with transcriptomic and proteomic experiments, to begin to understand the genes and protein products needed for aerial development and sporulation. Interestingly, the data from these experiments suggest that this developmental system may share properties with both *Streptomyces* and *Bacillus* sporulation – two very different bacterial sporulating systems.

The overall observations are very exciting. There are, however, a number of areas where there seem to have been some overinterpretation of the results, and where some additional experiments may help to further strengthen the conclusions. There is also A LOT of data presented, and many of the figures and tables would benefit from some additional information being provided so that readers can better understand and appreciate the data being presented.

Specific comments:

1) Are the spores produced by YIM 93972 more resistant to environmental stress (e.g. heat, dessication) than its substrate hyphae? Bacterial spores are not only reproductive cells, but also have stress-resistant properties. It would be useful to know if this were true of archaeal spores as well.

2) Lines 110-115: The data presented in the extended data (Figure 1) nicely show a cell wall protein preparation from YIM 93972 in Fig. 1d, and then describe an in silico western blot for two proteins. How any of these are connected to the S-layer is not clear from the information provided. Please provide an additional explanation in the legend to explain how these. Please also provide data/and explanation for how the authors know that their strain produces halorhodopsins. Were these identified in the cell wall (membrane?) preparations? Or was it obvious from the genome sequence? Regarding bacterioruberin – are the genes encoding this compound know? – and if yes, are there any obvious candidates present in the sequenced genome? (the phrase 'not producing' is not the same as 'does not have the machinery to encode'...)

3) Figure 2a: It isn't clear why all of these strains are deemed to exhibit morphological differentiation. There are definitely pigmentation differences, but only three of them seem to have an aerial hyphae-like appearance, based on the plate images. Showing plate pictures over time, or SEM images of cells having different morphologies would better support this conclusion. It appears

from the methods description, that differentiation is media dependent – at least for YIM 93972. It would be useful to mention this in the main text (this is also the case for *Streptomyces*).

4) Lines 162-163: Is there microscopic evidence to support the presence of hyphae of different size and shape? The length of hyphae would naturally differ, but there don't appear to be massive differences in the images shown in Fig. 1b.

5) Line 179-186: The presence of genes production of different energy-storage compounds is interesting. It would be worth noting that trehalose is often produced/used by sporulating *Streptomyces*, while dipicolinic acid is typically confined to the sporulating Firmicutes like *Bacillus*. It also isn't clear from the reference provided, that poly(R)-hydroxyalkanoic acid biosynthesis has any role in spore survival. Based on some of the genes identified as being upregulated, the sporulation process described here may interestingly combine features of *Streptomyces* development, with those of *Bacillus* endospore development (e.g. dipicolinic acid production). This may be worth highlighting.

6) Lines 205-207 and 213: While transcriptomic and proteomic analyses can provide information on whether a particular mutation might impact the transcript/protein levels of a particular gene and its product, they cannot definitively show that a mutation is responsible for a phenotype. Please rephrase to soften these sections, both around the conclusions that can be drawn from these experiments, and the tentative role of ORF_238 in morphological differentiation (while it looks promising, in the absence of any additional supporting genetic data, 'may be' seems more appropriate than 'is' with respect to the involvement of this gene in morphological differentiation).

7) Lines 234-243: While it is possible that the ORF_2669-2673 operon contributes to bialaphos uptake and submerged sporulation, the data presented do not allow for the exclusion of other possibilities. With respect to the bialaphos resistance observed for the mutant, there are many other possibilities that could explain this observation. E.g. the mutant may have a different surface architecture that precludes bialaphos uptake (e.g. other proteins may sequester it on the surface); or the mutant may fail to process bialaphos into its toxic form. With respect to the conclusion regarding submerged sporulation, all transcriptomic and proteomic experiments seemed to have been done with surface-grown cultures, so it isn't clear how the observations could be directly correlated with behaviours in liquid culture.

8) When complementing the *S. coelicolor* bldK mutant with ORF_2669-2673, what promoter was used to drive the expression of the operon? ermE*? Does the ORF_2669-2673 operon share the greatest similarity with the bldK locus in *S. coelicolor*? If not, does overexpressing the most similar operon (under the ermE* promoter) also complement the bald mutant phenotype?

Editorial comments:

1) A suggestion but not a requirement: The authors could consider expanding the manuscript into a more traditional Introduction/Results&Discussion/Methods kind of format, as there is currently a lot of useful information included in the extended and supplementary data.

2) Figure 1c: Note includes all abbreviations but that for PGS – would be useful to include this in the figure itself if wanting to keep the others within the figure. Alternatively, all could be removed and simply defined in the legend.

3) Extended Data Figure 1b: It is not clear what this is showing – the 'S' and 'F' dimensions are not defined, nor are any of the spots labelled. Please either describe and label, or remove.

4) Extended Data Figure 1c: The legend mentions circled spots, but there don't appear to be any circles indicated in the figure.

5) Supplementary Table 1: for *Natronomonas moolapensis* should the morphological description instead be 'pleomorphic' (not 'peomorphic')?

6) Line 120-122: Within the text, it states that the YIM 93972 chromosome possesses 2 rRNA

operons, but in Supplementary Table 2, it mentions 6 rRNA. Does this suggest 3 genes/operon? Consistency or additional explanation here would be useful.

7) The NTG-mutagenesis and mutant isolation are important experiments. It is not completely clear, however, how these experiments were conducted. Based on the data presented in Extended Figure 2a, there were 100's of mutants of interest identified after 1 generation. But then there were only a handful of interest after 4 generations. What happened between the different generations? Some additional information in both the methods and legend would be useful. Based on the text description in lines 192-201, it isn't clear whether any of the 8 mutants sequenced shared mutations in the same gene/locus. I admit to also having been very confused by the data presented in 2c and d – the strains are not clear – is W wild type? What is the difference between AH and SH in many extended data figures? (aerial hyphae and substrate hyphae? – if yes, please define in the legends). It would also be very helpful to have additional explanations accompanying the numbers associated with the right hand side of the table in 2c and with the different samples in 2d. How they are currently presented makes it very challenging for this reviewer to understand and appreciate the genetic changes being represented or reported on.

8) It is not clear what is being shown in Extended Figure 3A. Is this transcriptomic data? Or proteomic data? The title suggests transcriptomic, while the legend suggests proteome. Please expand the legend to ensure the figure can be clearly understood without any additional reference.

9) What are the numbers presented in Extended Tables 4, 5, 6? Ratio of what? Similarly, it is unclear what all of the values presented in Supplementary tables 5-8 and Extended data, Tables 4&5 represent. Please provide additional information within these tables.

10) Given the similarities between YIM 93972 development and *Streptomyces* development, it may be worth keeping in mind (and possibly mentioning within the text), that when aerial hyphae are being raised, there are still substrate hyphae present, so the later growth phases would represent a mixed population. If it appears that development proceeds differently in this archaeal species, it would be worth highlighting this.

11) The text in extended Figure 5 is too small to read without zooming in dramatically. Please consider using a larger font size.

12) Line 199: In bacteria, ParB is typically considered to be a DNA binding protein that promotes chromosome segregation, not a nuclease. Please check and adjust text if appropriate. In the future, it would be really interesting to probe how the multiple chromosomes/plasmids are properly segregated into the spores.

Reviewer #3 (Remarks to the Author):

The paper by Tang et al. reports the exciting description of a new species of halobacteria with a cellular differentiation similar to that of actinobacteria. It then provides molecular evidence for the role of specific genes in differentiation. The major concern is that the paper is so telescoped that it is not possible to evaluate the evidence for the molecular experiments. Moreover, many figures are presented without adequate explanation.

Overall, the paper would greatly benefit from separation into more than one manuscript. The description of the new species should be submitted to a systematics journal such as *Systematic and Applied Microbiology* or *International Journal of Systematic and Evolutionary Microbiology*. Properly written, it could also be justified in a higher impact and more general journal. However, it needs to address some of the critical factors discussed below.

The molecular or 'omics' experiments are likely to be very informative, but it is not possible to evaluate this presentation, esp. in the absence of methods. Of special concern, the data is presented without adequate explanation and it is not possible to actually see how the observations mentioned in the text are supported experimentally. For instance, Figure 3b and 3c report the

differential expression of genes and proteins, but there is no description of which conditions are being compared and which is the control condition. Likewise, in Figure 4, what are clusters 1, 2 and 3?

Specific comments:

Title: Why "Morphological cellular differentiation in a haloarchaeon". Cellular differentiation is usually morphological. Would "complex cellular differentiation in a haloarchaeon" be okay?

Line 55-57. This statement is factually incorrect, and it should be deleted or rewritten. Complex cellular differentiation has been reported in a number of archaea, including *Methanosarcina* and *Pyrodictium*. However, this does not detract from the importance of the work reported here.

Line 78. Please rewrite. The bacteria and archaea differ in many components of their 'information processing systems' in addition to translation. For instance, the enzymology of both transcription and DNA replication is also very different.

Line 90. Some authors might claim that cyst formation in *Methanosarcina mazei* is an example of complex cellular differentiation. This paragraph might be more useful to describe the cases of cellular differentiation in the archaea rather than cellular morphologies, which are not really the topic.

Lines 263-277. The authors are commended for naming their new and very interesting isolate. However, they failed to designate a type strain. According to the International Code of Nomenclature, the protologue reported here should designate a type strain and the accession numbers for it from two international culture collections from different countries. The genome assembly should also be designated here. It would also be valuable to report the criteria for assigning this as a novel genus and species. Typically, this would include low 16S rRNA sequence similarity and low Amino Acid Identity [AAI] to the most closely related halobacteria. See: Parker et al., *Int J Syst Evol Microbiol* 2019;69:S1

DOI 10.1099/ijsem.0.000778; see Rules 27 and 28.

Figure 1 is too complex and should be separated into individual components. 1a. Please indicate in the legend that each row indicates the days of incubation. How were the plates incubated? It probably isn't necessary to show more than one time point here anyway. 1b. There is no way to match the panels to the figure legend. Each panel should be labeled, and the labels should be referred to in the legend. 1c belongs in the discussion. 1d[2] is not labeled. What is this?

Figure 2. The colonial morphology of A0002 is not clear. Either include a clearer photograph or delete. On the basis of the phylogenetic tree, this new species could be classified in the genus *Halocatena*. Provide the rationale for classifying it in a novel genus. Usually this would require showing that the AAI is below a certain threshold. The phylogenetic tree might justify classification of YIM 93972 in a separate genus but not most of the other isolates.

Line 149. The halobacteria are now classified in the phylum Halobacteriota [see Rinke et al 2021: <https://doi.org/10.1038/s41564-021-00918-8>]

Figure 3b. The heat maps are too small to be read. Either they should not be presented or they should be presented in a format that is readable. The authors might consider only presenting the heatmaps for important sets of genes.

REVIEWER COMMENTS

Reviewer #1 (Remarks to the Author):

The authors report their isolation and identification of a halophilic archaeon exhibiting extensive morphological differentiation, including the presence of hyphae and spores. Multiple signature gene losses and gains are reported which may support the mycelial lifestyle of this organism. Transcriptomics and proteomics revealed a CDC48 ATPase (ORF_0238) may be involved in this morphological differentiation. Most interestingly, non-differentiating mutants obtained through chemical mutagenesis strongly implicated an oligopeptide transporter in areal hyphae differentiation. Specific comments:

1) Line 56: Complex morphological differentiation and pleomorphis has been well described in archaea previously (and quite complex, certainly at a molecular level), e.g. swimming or biofilm-related cell morphology transformations in haloarchaea, but what the authors might mean here is a metamorphosis/transformational type of differentiation, where the cell types are obviously and radically different in many different ways. This terminology should be corrected and could be made clearer and consistent throughout the paper. The authors should not use the term pleomorphology to describe their different cell types, because what they are claiming is a much more substantial state of cellular differentiation than how 'pleomorphology' is widely used in microbiology. (refer to textbook usages). Otherwise, the paper may attract additional criticism for claiming novelty that is not quite true. In this regard, the authors could consider revising the bold claim of the paper's title.

Response: We appreciate this advice and replaced this word with morphogenetic. The title of the manuscript has also been revised according to the suggestion from another reviewer.

2) Line 59: 'unlike other archaea' – the authors should at least qualify 'known archaea'.

Response: Yes indeed, the original phrase was inaccurate. It has been revised as suggested.

3) Line 136: The authors state on that when *H. pleomorpha* was plated onto ISP4 medium supplemented with NaCl, that similar morphological differentiation was observed to the newly isolated strains and that this may be common across Halobacteria (contradicting Line 59 comment). Is it possible that the chosen medium can induce morphological differentiation in many Halobacteria – majority of which are already described as pleomorphic (e.g. the model *H. salinarum*, *H. volcanii* etc)? Can the authors show evidence that such differentiation is limited to the phylogenetic branch implicated, as assumed?

Response: We collected *Haloferax volcanii* CGMCC 1.2150^T, *Halomarina orientis* JCM 16495^T, and *Haloplanus salinarum* JCM 31424^T and cultured them using ISP 4 medium. The results were included in the revised manuscript as Supplementary Fig. 2. None of these organisms showed morphological differentiation on ISP 4. Additionally, we cultured other six morphogenetic *halobacteria* using eight different media (Supplementary Fig. 1). These filamentous *halobacteria* have inconsistent morphological characteristics on different media. Therefore, these supplementary data seem to demonstrate that the chosen medium cannot induce morphological differentiation in other *halobacteria*.

4) How likely is it that these Haloarchaea (including YIM 93972) undergo morphological differentiation in their ecological environments and what benefits might this provide them with? This may be worth discussing briefly, or referring to environmental samples if possible, to show better evidence of the relevance of the different cell types.

Response: These strains exhibit previously unreported complex life cycles which do seem to be important for environmental adaptability. As other spore-forming strains (Nicholson W. L. *et al.* 2002), the spores of this new halophilic archaeon could be resistant to higher heat stress up to 80 °C than substrate hyphae (SH) up to 60 °C, which might enable them to

survive under extreme conditions by their protective and repair capabilities (Extended Data Fig. 1b). These spore-forming strains were isolated from high saline lakes owing to their ability to form highly resistant dormant spores. Their spores can rapidly convert from a dormant form in stress conditions to a fully active cell in nutrient rich environment. We added the related experiment and discussion in Page 5 Line 113-116.

Reference: Nicholson, W. L. et al. Bacterial endospores and their significance in stress resistance. *Antonie Van Leeuwenhoek* 81, 27-32 (2002).

5) Are the apparent spores shown to be dormant and non-replicative? Furthermore, do they show substantially elevated broad-spectrum stress resistance, as expected for the definition of a spore?

Response: We tested the temperature tolerance of the spores of YIM 93972 (please see Extended data Fig. 1b). The highest temperature that 93972 can tolerate is only 70 °C. So, the spore of YIM 93972 is different from the endospore of bacteria like *Bacillus*. This is mentioned in the revised manuscript.

6) Line 150: *Natronomonas* is not shown in Figure 2b as part of the sister clade of the pleomorphic clade but is referenced in the text.

Response: We are grateful for this comment. The text has been revised in Page 8 Line 187-191.

“Phylogenomic analysis of 268 low-paralogy haloCOGs that are universally conserved in 130 genomes of class *Halobacteria* (Supplementary Table 3) showed that YIM 93972 and its morphogenetic relatives including *H. pleomorpha*, formed a clade with *Halomarina orientis*, within the family *Halobacteriaceae* (Fig. 2b, Supplementary_data_file_2).”

7) Line 163: As noted, CetZ proteins are necessary for controlling pleomorphic shape changes in other *Halobacteria*. In addition, supplementary table 4C shows the loss of

arCOG06187 corresponding to a possible MreB (or actin-like) homologue. Why would the loss of these cell shape controlling proteins be associated with or promote the pleomorphic phenotype observed in the current study? Are any other cytoskeletal homologues present that may contribute to pleomorphism in these isolates?

Response: We appreciate this comment that led us to make an additional, potentially interesting observation. CetZ is involved in maintaining the rod-like cell shape in non-pleomorphic archaea, such as *Haloferax volcanii*. Considering that pleomorphic *Haloarchaea* do not have to maintain the rod shape, the loss of CetZ appears to be consistent with their phenotype, and moreover could facilitate differentiation. Clearly, this is only an inference based on the absence of this gene in all pleomorphic strains. Accordingly, we added the following in Page 9 Line 208-212: “In non-morphogenetic halobacteria, in particular, *Haloferax volcanii*, CetZ is involved in the maintenance of the rod-like cell shape (Duggin, I. G. *et al.* 2015). The morphogenetic haloarchaea do not form rod-shaped cells, and rod shape might be incompatible with cell pleomorphism, suggesting a functional link between the loss of *cetZ* and cellular differentiation.” The situation with the MreB/Mbl family is more complicated. *Haloarchaea* do not encode a bona fide MreB which in bacteria is involved in cell rod-shape determination. In archaea, the bona fide MreB is present only in several Methanogens that have a peptidoglycan cell wall (Ithurbide S, *et al.* 2022). However, *Haloarchaea* encode two MreB-like protein families (mostly in two haloCOGs: halo.02581 and halo.02163), the function of which is not known. Prompted by the reviewer’s question, we constructed phylogenetic tree of these families using MreB orthologs from methanogens and several other archaea as an outgroup and found a specific amplification of halo.02163 in the pleomorphic haloarchaea. We now show this tree and phyletic pattern of two haloCOGs as a new supplementary figure and mention this observation in the text as follows: “Conversely, in pleomorphic haloarchaea, the halo.02163 family of MreB-like proteins is expanded, whereas members of the other MreB-like family that is common in other haloarchaea (halo.02581) is lost. Given that MreB is the key cell shape determinant in bacteria (Shi, H. *et al.* 2018), these findings further

emphasize specific changes in cytoskeleton organization that are likely to be relevant for the pleomorphism of these archaea (Extended Data Fig. 6”).

We believe that experimental study of cytoskeleton in pleomorphic Haloarchaea is a research direction of great promise, especially, when and if genetic tools are established for these organisms.

Reference:

Duggin, I. G. et al. CetZ tubulin-like proteins control archaeal cell shape. *Nature* 519, 362-365 (2015).

Ithurbide, S. et al. Spotlight on FtsZ-based cell division in Archaea. *Trends in Microbiology* 30, 665-678 (2022)

Shi, H. et al. How to build a bacterial cell: MreB as the foreman of *E. coli* construction. *Cell*, 172(6), 1294-1305 (2018).

8) Line 165: The type IV pili system typified by the PilB3 locus of *H. volcanii* is present outside of Halobacteria as well, including Methanomicrobia and Archaeoglobi according to Makarova et al., 2016 “Diversity and Evolution of Type IV pili Systems in Archaea”, supplementary Figure 3.

Response: According to Supplementary Figure S3 in Makarova et al., 2016 “Diversity and Evolution of Type IV pili Systems in Archaea”, PilB3-like systems are present in Halobacteria and Methanomicrobia only. Thus, we corrected the text in Page 9 Line 204-207 as follows: “These include the loss of CetZ (FtsZ3) which is involved in maintaining cell shape in *Halobacteria* (Duggin, I. G. et al. 2015), the archaeellum and another halobacterial type IV pili system, typified by the PilB3 locus of *Haloferax volcanii* (Duggin, I. G. et al. 2015), which is represented in *Halobacteria* and *Methanomicrobia* (Makarova, K. S. et al. 2016).”.

Reference:

Duggin, I. G. et al. CetZ tubulin-like proteins control archaeal cell shape. *Nature* 519, 362-365 (2015).

Makarova, K. S., Koonin, E. V. & Albers, S. V. Diversity and Evolution of Type IV pili Systems in Archaea. *Front. Microbiol.* 7, 667 (2016).

9) Line 169: Are there any examples in the literature that CDC48 family ATPases contribute specifically to pleomorphism or shape changes including hyphae formation or sporulation in other organisms?

Response: No, we are unaware of such experimental data. Nevertheless, based on the data presented in this work (both in silico and experimental) and the general role of CDC48 ATPases as regulators of key cellular processes (Barthelme, D. *et al.* 2016), we believe it is a strong candidate for experimental follow up to determine its role in pleomorphism. That said, the inference was made more cautious in the revised manuscript.

Reference:

Barthelme, D. & Sauer, R. T. Origin and Functional Evolution of the Cdc48/p97/VCP AAA+ Protein Unfolding and Remodeling Machine. *J. Mol. Biol.* 428, 1861-1869 (2016).

10) Line 213: The authors defined the bald mutants as having no aerial hyphae but do not suggest or show that these mutants are also sporulation deficient. I agree that ORF_0238 could potentially be important for morphological differentiation generally, but its role may be direct or indirect and having knock-on affects on morphological differentiation. The conclusion that ORF_0238 itself is functionally important for morphological differentiation of hyphae and sporulation may be too strong given the evidence – especially if there are no specific examples of CDC48 family ATPases contributing to pleomorphism, morphological differentiation, or sporulation in the literature. If there are homologues of ORF_0238 in model bacteria or archaea it may be informative to conduct complementation studies similar to those in Figure 5 to strengthen the evidence.

Response: Multiple successive passage experiments confirmed that bald mutants are sporulation deficient. We believe ORF_0238 was one of important factors for

morphological differentiation of hyphae and sporulation. Per your suggestion, we performed homology analysis in spore-forming Actinobacteria and bacillus by microbiologist, and did not identify any orthologs of ORF_238 (there are, obviously, many other AAA+ ATPases). Therefore, an appropriate complementation experiment cannot be designed and performed.

11) Line 247 and 497: Complementation of the *S. coelicolor* M145 oligopeptide permease deletion with both YIM 93972 ORF_2669-2673 and the endogenous *S. coelicolor* oligopeptide permease does appear to restore mycelia and spores, however the morphology of the spores in the complementation strains look potentially larger and less round or differentiated compared to M145/pIB139, perhaps indicating only partial complementation rather than full restoration of the sporulation defect.

Response: This complementation experiment was performed in the pKCCas9dO plasmid but not in situ. There were differences in foreign plasmid DNA and in situ for the gene cluster ORF_2669-2673. In addition, the homologous gene cluster, ORF_2669-2673 of YIM 93972, has rather distant phylogenetic relationships with *bldKA-KE* of *S. coelicolor* M145.

Reviewer #2 (Remarks to the Author):

The fascinating study of Tang et al. report a novel mode of development for a haloarchaeal species that physically resembles the classical sporulating life cycle of *Streptomyces* bacteria. The sporulating properties of this organism seem confined to a specific clade within the Halobacteria, and seem to be associated with genes unique to these bacteria. The absence of genetic tools for this organism precluded any in-depth genetic analyses, but mutational studies successfully yielded mutants defective in their ability to raise aerial hyphae. The authors followed up these mutational analyses with transcriptomic and proteomic experiments, to begin to understand the genes and protein products needed for aerial development and sporulation. Interestingly, the data from these experiments suggest that this developmental system might share properties with both *Streptomyces* and *Bacillus* sporulation – two very different bacterial sporulating systems.

The overall observations are very exciting. There are, however, a number of areas where there seem to have been some overinterpretation of the results, and where some additional experiments may help to further strengthen the conclusions. There is also A LOT of data presented, and many of the figures and tables would benefit from some additional information being provided so that readers can better understand and appreciate the data being presented.

Specific comments:

1) Are the spores produced by YIM 93972 more resistant to environmental stress (e.g. heat, dessication) than its substrate hyphae? Bacterial spores are not only reproductive cells, but also have stress-resistant properties. It would be useful to know if this were true of archaeal spores as well.

Response: We tested the temperature tolerance of the spores of YIM 93972 (please see Extended data Fig. 1b). The highest temperature that 93972 can tolerate is only 70 °C. So,

the spore of YIM 93972 is different from the endospore of bacteria like *Bacillus*. We added the related experiment and discussion in Page 5 Line 113-116.

As other spore-forming strains (Nicholson W. L. *et al.* 2002), the spores of this new halophilic archaeon could resistant to higher heat stress up to 80 °C than substrate hyphae (SH) up to 60 °C, which might enable them to survive under extreme conditions by their protective and repair capabilities (Extended Data Fig. 1b).

Reference: Nicholson, W. L. *et al.* Bacterial endospores and their significance in stress resistance. *Antonie Van Leeuwenhoek* 81, 27-32 (2002).

2) Lines 110-115: The data presented in the extended data (Figure 1) nicely show a cell wall protein preparation from YIM 93972 in Fig. 1d, and then describe an in silico western blot for two proteins. How any of these are connected to the S-layer is not clear from the information provided. Please provide an additional explanation in the legend to explain how these. Please also provide data/and explanation for how the authors know that their strain produces halorhodopsins. Were these identified in the cell wall (membrane?) preparations? Or was it obvious from the genome sequence? Regarding bacterioruberin – are the genes encoding this compound know? – and if yes, are there any obvious candidates present in the sequenced genome? (the phrase ‘not producing’ is not the same as ‘does not have the machinery to encode’...)

Response: Besides the evidence from the genome sequence (halorhodopsin gene ORF_1384, halo.02535), Production of halorhodopsin by YIM 93972 was also confirmed by in silico Western blot and the data of LC-MS (Extended Data Fig. 3). The key gene involved in bacterioruberin biosynthesis, lycopene elongase (Peck, R. F. *et al.*, 2017), belongs to halo.01227. These genes are absent in YIM 93972 and other morphogenetic Halobacteria, in contrast to halorhodopsin gene (halo.02535) which is encoded in all these genomes. The text has been modified accordingly.

Reference: Peck, R. F. *et al.* Opsin-mediated inhibition of bacterioruberin synthesis in halophilic archaea. *Journal of bacteriology*, 199(21), e00303-17 (2017).

As for S-layer, two proteins identified in our screen, ORF_1400 and ORF_1704 (halo.06692 and halo.10677 respectively) belong to S-layer-forming halobacterial major cell surface glycoprotein family (arCOG10180) annotated based on sequence similarity with Csg, HVO_2072. The latter protein, however, is only distantly related and belongs to arCOG03906 family, orthologs of which are missing in YIM 93972. The S-layer proteins are not highly conserved and arCOG03906 are missing in many Halobacteria. Similarly to csg, these are secreted proteins with characteristic PGF-CTERM archaeal protein-sorting signal (Abdul Halim, M. F. *et al.* 2013). Taken together, both *in silico* and experimental data indicate that these are the most likely S-layer proteins in YIM 93972. These details are included in the revised manuscript.

Reference: Abdul Halim, M. F. *et al.* Haloferax volcanii archaeosortase is required for motility, mating, and C-terminal processing of the S-layer glycoprotein. *Molecular microbiology*, 88(6), 1164-1175 (2013).

3) Figure 2a: It isn't clear why all of these strains are deemed to exhibit morphological differentiation. There are definitely pigmentation differences, but only three of them seem to have an aerial hyphae-like appearance, based on the plate images. Showing plate pictures over time, or SEM images of cells having different morphologies would better support this conclusion. It appears from the methods description, that differentiation is media dependent – at least for YIM 93972. It would be useful to mention this in the main text (this is also the case for Streptomyces).

Response: According to this comment, we recultivated these seven morphogenetic halobacteria and updated the images of colony morphology. Additionally, the SEM images were also be appended (Figure 2a). Although the growth of the seven strains was different, the aerial hyphae were clearly discernible.

4) Lines 162-163: Is there microscopic evidence to support the presence of hyphae of

different size and shape? The length of hyphae would naturally differ, but there don't appear to be massive differences in the images shown in Fig. 1b.

Response: We are grateful for this comment. The text has been revised accordingly.

5) Line 179-186: The presence of genes production of different energy-storage compounds is interesting. It would be worth noting that trehalose is often produced/used by sporulating *Streptomyces*, while dipicolinic acid is typically confined to the sporulating Firmicutes like *Bacillus*. It also isn't clear from the reference provided, that poly(R)-hydroxyalkanoic acid biosynthesis has any role in spore survival. Based on some of the genes identified as being upregulated, the sporulation process described here may interestingly combine features of *Streptomyces* development, with those of *Bacillus* endospore development (e.g. dipicolinic acid production). This may be worth highlighting.

Response: All these genes are common in non-pleomorphic Halobacteria, too (for example, Poly(3-hydroxyalkanoate) synthetase - arCOG06344, 4-hydroxy-tetrahydrodipicolinate synthase - arCOG04172, Trehalose-6-phosphate synthase - arCOG02831 in the Supplementary Table 3). Moreover, the best hits for all these proteins are also Halobacterial. Considering that only poly(R)-hydroxyalkanoic acid biosynthesis genes are strongly upregulated in sporulating hyphae, we hypothesize that this is a major storage molecule in the spores. Taking this into account we made the following modifications in Page 10-11 Line 240-251:

“Sporulating bacteria employ a distinct set of small molecules for energy storage and spore protection. We identified genes present in all 7 genomes of morphogenetic haloarchaea that are involved in trehalose biosynthesis and utilization²⁸, dipicolinic acid biosynthesis²⁹ and poly(R)-hydroxyalkanoic acid biosynthesis³⁰ (Supplementary Table 3). Comparative omics data indicated that genes involved in the biosynthesis of small molecules implicated in sporulation, namely, Poly(R)-hydroxyalkanoic acid synthesis (*ORF_2608*) and 4-hydroxy-tetrahydrodipicolinate (*ORF_1392*), were upregulated 3.59 and 7.83 folds at the

mRNA, and 1.31 and 1.57-fold at the protein level, respectively. This observation suggests that poly(R)-hydroxyalkanoic acid could be a key storage molecule in the spores. Polyhydroxyalkanoates also accumulate in hyphae and spores in some actinobacterial species³⁰. However, all these genes are also widespread in non-morphogenetic *Halobacteria*.”

6) Lines 205-207 and 213: While transcriptomic and proteomic analyses can provide information on whether a particular mutation might impact the transcript/protein levels of a particular gene and its product, they cannot definitively show that a mutation is responsible for a phenotype. Please rephrase to soften these sections, both around the conclusions that can be drawn from these experiments, and the tentative role of ORF_238 in morphological differentiation (while it looks promising, in the absence of any additional supporting genetic data, ‘may be’ seems more appropriate than ‘is’ with respect to the involvement of this gene in morphological differentiation).

Response: We are grateful for this comment. The text has been revised to soften the claims.

7) Lines 234-243: While it is possible that the ORF_2669-2673 operon contributes to bialaphos uptake and submerged sporulation, the data presented do not allow for the exclusion of other possibilities. With respect to the bialaphos resistance observed for the mutant, there are many other possibilities that could explain this observation. E.g. the mutant may have a different surface architecture that precludes bialaphos uptake (e.g. other proteins may sequester it on the surface); or the mutant may fail to process bialaphos into its toxic form. With respect to the conclusion regarding submerged sporulation, all transcriptomic and proteomic experiments seemed to have been done with surface-grown cultures, so it isn’t clear how the observations could be directly correlated with behaviours in liquid culture.

Response: We appreciate this advice. To observe the bialaphos resistance, liquid culture experiment was performed, and the results were consistent with those obtained on solid cultures.

8) When complementing the *S. coelicolor* *bldK* mutant with ORF_2669-2673, what promoter was used to drive the expression of the operon? *ermE**? Does the ORF_2669-2673 operon share the greatest similarity with the *bldK* locus in *S. coelicolor*? If not, does overexpressing the most similar operon (under the *ermE** promoter) also complement the bald mutant phenotype?

Response: Considering the relatively large intergenic region between ORF_2669 and ORF_2670, we added the operon *ermEp** before ORF_2669 and ORF_2670, respectively. Yes, ORF_2669-2673 operon shares the greatest similarity with the *bldK* locus in *S. coelicolor*.

Editorial comments:

1) A suggestion but not a requirement: The authors could consider expanding the manuscript into a more traditional Introduction/Results&Discussion/Methods kind of format, as there is currently a lot of useful information included in the extended and supplementary data.

Response: The format of manuscript has been revised accordingly.

2) Figure 1c: Note includes all abbreviations but that for PGS – would be useful to include this in the figure itself if wanting to keep the others within the figure. Alternatively, all could be removed and simply defined in the legend.

Response: all abbreviations appeared in Figure 1 have been reorganized in the legend of Figure 1.

3) Extended Data Figure 1b: It is not clear what this is showing – the ‘S’ and ‘F’ dimensions are not defined, nor are any of the spots labelled. Please either describe and label, or remove.

Response: The Extended Data Figure 1 and the corresponding legend have been revised.

4) Extended Data Figure 1c: The legend mentions circled spots, but there don't appear to be any circles indicated in the figure.

Response: The Extended Data Figure 1 and the corresponding legend have been revised.

5) Supplementary Table 1: for *Natronomonas moolapensis* should the morphological description instead be ‘pleomorphic’ (not ‘peomorphic’)?

Response: We are grateful for this comment. This mistake has been corrected.

6) Line 120-122: Within the text, it states that the YIM 93972 chromosome possesses 2 rRNA operons, but in Supplementary Table 2, it mentions 6 rRNA. Does this suggest 3 genes/operon? Consistency or additional explanation here would be useful.

Response: We are grateful for this comment. The title of the corresponding column in Supplementary Table 2 has been revised to ‘rRNA gene’.

7) The NTG-mutagenesis and mutant isolation are important experiments. It is not completely clear, however, how these experiments were conducted. Based on the data presented in Extended Figure 2a, there were 100's of mutants of interest identified after 1 generation. But then there were only a handful of interest after 4 generations. What happened between the different generations? Some additional information in both the

methods and legend would be useful. Based on the text description in lines 192-201, it isn't clear whether any of the 8 mutants sequenced shared mutations in the same gene/locus. I admit to also having been very confused by the data presented in 2c and d – the strains are not clear – is W wild type? What is the difference between AH and SH in many extended data figures? (aerial hyphae and substrate hyphae? – if yes, please define in the legends). It would also be very helpful to have additional explanations accompanying the numbers associated with the right hand side of the table in 2c and with the different samples in 2d. How they are currently presented makes it very challenging for this reviewer to understand and appreciate the genetic changes being represented or reported on.

Response: We performed a four-generation mutant screening. The morphological phenotype of many mutants recovered in the next sub-cultivation. Therefore, after four generations, only 5 transitional mutants and 3 bald mutants remained with stable morphological mutant phenotypes. The additional explanation has been added in both methods and legend of Extended Data Figure 8 (original Extended Figure 2). To the text description in (original) lines 192-201, we made the following modifications in Page 11 Line 263-266: “Analysis of the genome sequences of these 8 mutants identified two mutations in protein-coding gene sequences occurred in three bald mutants; one mutation upstream of a gene occurred in three bald mutants; and one mutation upstream of a gene in the three of five transitional mutants (Extended Data Fig. 8c, Extended data Table 1).” The additional information about abbreviations (W, wild type; T, transitional mutant; B, bald mutant; AH, aerial hyphae; SH, substrate hyphae.) has been added in the legend of related figures. And, the numbers appeared in tables c and d are the number of reads in re-sequencing. So, the panels c and d have been revised to make the meaning of numbers in the right-hand side of tables c and d clearer.

8) It is not clear what is being shown in Extended Figure 3A. Is this transcriptomic data? Or proteomic data? The title suggests transcriptomic, while the legend suggests proteome. Please expand the legend to ensure the figure can be clearly understood without any

additional reference.

Response: Yes, this is transcriptomic data. The original Extended Figure 3 has been reorganized in Figure 3 (b and d). The related legend has also been revised.

9) What are the numbers presented in Extended Tables 4, 5, 6? Ratio of what? Similarly, it is unclear what all of the values presented in Supplementary tables 5-8 and Extended data, Tables 4&5 represent. Please provide additional information within these tables.

Response: We appreciate this helpful advice. We added the detailed information of each file in our revised material. W-SH was the control sample. Genes with ratios greater than 1.5-fold and p -value smaller than 0.05 (Student's t-test) were considered as regulated. Protein changes greater than 1.5-fold and p -value smaller than 0.05 (Significance A) were considered as regulated. W-SH, wild substrate hyphae; W-AH, wild type aerial hyphae; T-SH, transitional substrate hyphae; B-SH, bald substrate hyphae. T_, transcriptomics; P_, proteomics.

10) Given the similarities between YIM 93972 development and *Streptomyces* development, it may be worth keeping in mind (and possibly mentioning within the text), that when aerial hyphae are being raised, there are still substrate hyphae present, so the later growth phases would represent a mixed population. If it appears that development proceeds differently in this archaeal species, it would be worth highlighting this.

Response: The cellular development of this new haloarchaeon includes the growth of substrate and aerial hyphae and the spore formation, which resembles the morphogenetic development of *Streptomyces*. This distinguishing feature is mentioned in the revised text.

11) The text in extended Figure 5 is too small to read without zooming in dramatically. Please consider using a larger font size.

Response: Extended data Figure 5 has been changed to Extended data Figure 9. The quality and size of the pictures has also been improved.

12) Line 199: In bacteria, ParB is typically considered to be a DNA binding protein that promotes chromosome segregation, not a nuclease. Please check and adjust text if appropriate. In the future, it would be really interesting to probe how the multiple chromosomes/plasmids are properly segregated into the spores.

Response: Yes, indeed, this is a DNA binding protein. Corrected.

Reviewer #3 (Remarks to the Author):

1) The paper by Tang et al. reports the exciting description of a new species of halobacteria with a cellular differentiation similar to that of actinobacteria. It then provides molecular evidence for the role of specific genes in differentiation. The major concern is that the paper is so telescoped that it is not possible to evaluate the evidence for the molecular experiments. Moreover, many figures are presented without adequate explanation.

Overall, the paper would greatly benefit from separation into more than one manuscript. The description of the new species should be submitted to a systematics journal such as Systematic and Applied Microbiology or International Journal of Systematic and Evolutionary Microbiology. Properly written, it could also be justified in a higher impact and more general journal. However, it needs to address some of the critical factors discussed below.

Response: According to the requirements for the format of article, the manuscript has been revised thoroughly and expanded. In the revised version, a new subsection of results was organized for the taxonomy of this new species. In this subsection, we provided the evidence that support the proposal of new genus.

2) The molecular or 'omics' experiments are likely to be very informative, but it is not possible to evaluate this presentation, esp. in the absence of methods. Of special concern, the data is presented without adequate explanation and it is not possible to actually see how the observations mentioned in the text are supported experimentally. For instance, Figure 3b and 3c report the differential expression of genes and proteins, but there is no description of which conditions are being compared and which is the control condition. Likewise, in Figure 4, what are clusters 1, 2 and 3?

Response: In the revised version, the transcriptomic and proteomic results were separated (Figure 3 and 4). And the treatment and control conditions used for

transcriptomic and proteomic comparison were clearly presented in Figure 3e and Figure 4f. In all ratio calculation, W-SH was the control sample. In transcriptomic experiment, T1-T2 are two biological replicates of transitional mutants. B1-B3 are three biological replicates of bald mutants. Each sample has three technical replicates. Genes with ratios greater than 1.5-fold and *p*-value smaller than 0.05 (Student's *t*-test) were considered as regulated. The expression of dysregulated genes was clustered with 9 subgroups by Mfuzz (Futschik, M. E., *et al*, 2005) using R software (Fig. 3d).

Reference: Futschik, M. E., *et al*. Noise-robust soft clustering of gene expression time-course data. *Journal of bioinformatics and computational biology*, 3(04), 965-988 (2005).

In proteomic experiments, T1-T2 are two biological replicates of transitional mutants. B1-B3 are three biological replicates of bald mutants. W-SH1 and W-SH2, T1-SH1 and T1-SH2, B2-SH1 and B2-SH2, are two biological replicates from wild, transitional, and bald group, respectively. Protein changes greater than 1.5-fold and *p*-value smaller than 0.05 (Significance A, Cox, J. *et al*. 2008) were considered as regulated. The expression of dysregulated genes was clustered with 9 subgroups by R software (Fig. 4e).

In new revised Fig. 5a, we listed the significant consistently expression patterns (Cluster 1, 2, and 3) of the substrate hyphae (SH) of mutated strain (T or B) vs. substrate hyphae (SH) of wild (W) strain, and aerial hyphae (AH) of wild strain vs. substrate hyphae (SH) of wild strain.

Reference: Cox, J., & Mann, M. MaxQuant enables high peptide identification rates, individualized ppb-range mass accuracies and proteome-wide protein quantification. *Nature biotechnology*, 26(12), 1367-1372 (2008).

Specific comments:

3) Title: Why “Morphological cellular differentiation in a haloarchaeon”. Cellular differentiation is usually morphological. Would “complex cellular differentiation in a haloarchaeon” be okay?

Response: We agree, this is an excellent point. Changed accordingly throughout.

4) Line 55-57. This statement is factually incorrect, and it should be deleted or rewritten. Complex cellular differentiation has been reported in a number of archaea, including *Methanosarcina* and *Pyrodictium*. However, this does not detract from the importance of the work reported here.

Response: We appreciate this comment. In the Abstract, there is no room for specifics, but changed to “However, there are few known cases of complex cell differentiation in archaea.”

5) Line 78. Please rewrite. The bacteria and archaea differ in many components of their ‘information processing systems’ in addition to translation. For instance, the enzymology of both transcription and DNA replication is also very different.

Response: We believe the sentence is largely accurate as written. There is no claim that differences in the translation are the only ones between archaea and bacteria, only that most universal phylogenies include translation system components. We nevertheless modified and amended the sentence as follows “Apart from the sharp separation in the phylogenetic trees of universal genes (mostly encoding translation system components), archaea differ from bacteria in many major features including partly unrelated DNA replication and transcription machineries, different structures of membrane lipids and cell walls, and the corresponding, distinct enzymatic machineries involved in membrane and cell wall biogenesis, several unique coenzymes, and unique RNA modifications.”

6) Line 90. Some authors might claim that cyst formation in *Methanosarcina mazei* is an example of complex cellular differentiation. This paragraph might be more useful to describe the cases of cellular differentiation in the archaea rather than cellular morphologies, which are not really the topic.

Response: Some authors might make such claims, and so we mention this in the revised manuscript, but still, this is hardly an example of complex differentiation, as pointed out in the revision.

7) Lines 263-277. The authors are commended for naming their new and very interesting isolate. However, they failed to designate a type strain. According to the International Code of Nomenclature, the protologue reported here should designate a type strain and the accession numbers for it from two international culture collections from different countries. The genome assembly should also be designated here. It would also be valuable to report the criteria for assigning this as a novel genus and species. Typically, this would include low 16S rRNA sequence similarity and low Amino Acid Identity [AAI] to the most closely related halobacteria. See: Parker et al., *Int J Syst Evol Microbiol* 2019;69:S1 DOI 10.1099/ijsem.0.000778; see Rules 27 and 28.

Response: The taxonomic descriptions has been revised. Discussion of the evidence in support of the proposal of new genus also was added in the revised manuscript.

8) Figure 1 is too complex and should be separated into individual components. 1a. Please indicate in the legend that each row indicates the days of incubation. How were the plates incubated? It probably isn't necessary to show more than one time point here anyway. 1b. There is no way to match the panels to the figure legend. Each panel should be labeled, and the labels should be referred to in the legend. 1c belongs in the discussion. 1d[2] is not labeled. What is this?

Response: Figure 1 has been reorganized, and all subgraphs have also labeled. And, about the 1c (changed to 1e) of figure 1, our idea is to visually summarize and show the similarities and differences between YIM 93972 and the bacteria and archaea in terms of phenotypic data.

9) Figure 2. The colonial morphology of A00012 is not clear. Either include a clearer photograph or delete. On the basis of the phylogenetic tree, this new species could be classified in the genus *Halocatena*. Provide the rationale for classifying it in a novel genus. Usually this would require showing that the AAI is below a certain threshold. The phylogenetic tree might justify classification of YIM 93972 in a separate genus but not most of the other isolates.

Response: We recultivated these seven morphogenetic halobacteria and updated the images of colony morphology. Additionally, the SEM images were also be appended (Figure 2). In the section on taxonomic description, the evidence supporting the proposal of new genus is included.

10) Line 149. The halobacteria are now classified in the phylum Halobacteriota [see Rinke et al 2021: <https://doi.org/10.1038/s41564-021-00918-8>]

Response: Yes, the new phylum name *Halobacteriota* has been proposed, but has not been ratified (announcement in Validation Lists). Therefore, in order to avoid any misunderstanding, we only mention the family name here in the revised version of manuscript.

11) Figure 3b. The heat maps are too small to be read. Either they should not be presented or they should be presented in a format that is readable. The authors might consider only presenting the heatmaps for important sets of genes.

Response: the original figure 3 has been separated into new Figure 3 and 4 for presenting transcriptomic and proteomic data respectively.

Reviewer #1 (Remarks to the Author):

The additional experiments and revisions by Tang et al have have done a good job to address most the previous concerns satisfactorily. There are some remaining points on the revised version of the text to be addressed.

1. The title can be improved as it is broad sweeping and not really accurate because a strong case can still be made that this not the first example of complex cellular differentiation in a haloarchaeon. The title should properly define the paper's main novel finding. E.g. Include aerial hyphae or spore terms? For example, the running title is short yet has more precise information than the main title...
2. The text in extended data figure 3c, extended data figure 6b, extended data figure 7c and d is too small to read.
3. Line 199 and below: The statement that CetZ is 'maintaining' cell shape is incorrect. It does the opposite. Please change this word to 'changing' or 'controlling'.
4. Line 202-204 and this paragraph: This section and the statement "rod shape might be incompatible with cell pleomorphism" are incorrect and confusing. *H. volcanii* and other related archaea are widely acknowledged to be pleomorphic, and also differentiate into motile rods. This would also be understood by readers to be a type of 'morphogenetic' differentiation – just a different type compared to the particular type described by the authors. The paper should be revised throughout to clarify. Here, the exact type of morphological response needs to be specified while referring to the evidence.
5. Line 205. The morphology change associated with *H. volcanii* motile rod differentiation is known to be dependent on CetZ1. The authors effectively state the opposite, so please correct this. Again, always specify what type of differentiation is being described ('morphogenetic' is not specific enough to define the type described here, and would certainly cause confusion for readers in future).
6. There is also no logical basis given for the claim that rod shape may be incompatible with the observed specific type of morphogenesis/cellular differentiation (not pleomorphism generally) in the current species. The authors do not see rod development in their species described here, but that does not mean they cannot form rods under other conditions. I would recommend keeping these statements more open to potential discoveries to the contrary in future.

Reviewer #2 (Remarks to the Author):

This is exciting work, and in this revised iteration the authors have done a great job of addressing the comments provided by the previous reviewers. I had only relatively minor comments for the authors to consider.

- 1) Lines 112-113 and Figure 1b: Spore chains are beautifully evident from the solid culture samples, but they are not as obvious from the liquid-grown culture
- 2) Extended figure 1b – growth (presumably lawns?) is not all that obvious on many of the plates grown at lower temperatures
- 3) Supplementary plate pictures – it is often challenging to see growth relative to the background plate (Figures 1,2)
- 4) Lines 253-254: It is not clear why 'immunity' is relevant and connected with the morphogenic characteristics being described in this section?
- 5) Line 315: it isn't clear why it is being proposed that the oligopeptide permease may only be involved in submerged sporulation (and not also during growth on solid substrates)? Was bialaphos resistance tested only during liquid culture? – Figure 5d suggests it was done on solid plates. There is no information about how this testing was done in the methods section of the manuscript – it would be helpful to include this information.
- 6) Lines 324-336: For this section, actinobacteria and cyanobacteria are highlighted, but then all

examples discussed are from *Bacillus subtilis*. It may be worth including the sporulating Firmicutes in the opening sentence, and then also making clear that the YlbF, BoIA and AbrB-family regulators have all been characterized in *Bacillus subtilis*.

7) Line 664: This would be a negative control, not a positive control.

8) Do the transitional mutants shown in Fig. 5c grow better than wild type or the bald mutants? Growth seems to be much more dense for this mutant than any of the other strains, both in the presence and absence of bialaphos. This could be worth commenting on.

Reviewer #3 (Remarks to the Author):

While the manuscript is greatly improved, the presentation would benefit from careful copyediting and expanding more details in the methods section. Suggestions are included on the attached pdf to help in the revision.

In general, the reader should be able to understand the figures from the legend without having to refer to the methods and results sections.

The methods should be described in enough detail to allow another lab to reproduce them, especially for those involving the biological materials, ie. isolation of spores, etc.

Extended Data Fig. 8 | The isolation and characteristics of YIM 93972 mutants panels c and d. What does it mean when the mutation includes the entire coding region, ie. such as T4, last column.

Supplementary Fig. 3 | The analysis of quantitative proteomics by TMT-labeling. The point of c is not clear. What do the fraction numbers correspond to?

In their review of the second version of this manuscript, reviewer #3 added some comments to the manuscript file. These comments were forwarded to the authors, who replied as included in this Peer Review File.

Enclosed please find the revised manuscript. We are pleased to know that all three reviewers generally agreed to accept our revised manuscript. We highly appreciate the insightful comments from the editor and reviewers. We revised our manuscript again by following the editor and reviewers' suggestions. The point-by-point response is included in the rebuttal letter.

Thank you very much for handling our manuscript!

Sincerely,

Ping Xu, Eugene Koonin, Xiaoyang Zhi, and Yao Zhang

February 20, 2023

Manuscript ID: NCOMMS-22-23864-T

Title: " Morphological cellular differentiation in a haloarchaeon"

Enclosed please find the revised manuscript. We have been pleased to know that all three reviewers have recognized the impact of our work. We highly appreciate the insightful comments from the reviewers. We revised the manuscript again by following the reviewers' suggestions. The point-by-point response is as follows.

REVIEWER COMMENTS

Reviewer #1 (Remarks to the Author):

The additional experiments and revisions by Tang et al have have done a good job to address most the previous concerns satisfactorily. There are some remaining points on the revised version of the text to be addressed.

1. The title can be improved as it is broad sweeping and not really accurate because a strong case can still be made that this not the first example of complex cellular differentiation in a haloarchaeon. The title should properly define the paper's main novel finding. E.g. Include aerial hyphae or spore terms? For example, the running title is short yet has more precise information than the main title...

Response: We appreciate this advice. The title has been revised as "Cellular differentiation into hyphae and spores in halophilic archaea", and the running title has been revised as "A new morphogenetic type of haloarchaea".

2. The text in extended data figure 3c, extended data figure 6b, extended data figure 7c and d is too small to read.

Response: These figures have been edited.

3. Line 199 and below: The statement that CetZ is 'maintaining' cell shape is incorrect. It does the opposite. Please change this word to 'changing' or 'controlling'.

Response: We are very appreciated for this comment. The mistake has been corrected according to the suggestion.

4. Line 202-204 and this paragraph: This section and the statement "rod shape might be incompatible with cell pleomorphism" are incorrect and confusing. *H. volcanii* and other related archaea are widely acknowledged to be pleomorphic, and also differentiate into motile rods. This would also be understood by readers to be a type of 'morphogenetic' differentiation – just a different type compared to the particular type described by the authors. The paper should be revised throughout to clarify. Here, the exact type of morphological response needs to be specified while referring to the evidence.

Response: Yes, this new haloarchaeon just display a new type of morphogenetic differentiation, which is with hyphae differentiation and spore formation. So, we added a note when "morphogenetic haloarchaea" first appeared (Line 175-176): "morphogenetic haloarchaea (referring specifically to haloarchaea with hyphae differentiation and spore formation)". The whole sentence is "To gain further insight into the biology of morphogenetic haloarchaea (referring specifically to haloarchaea with hyphae differentiation

and spore formation), the complete genomes of YIM 93972 and other five morphogenetic strains were sequenced.”.

5. Line 205. The morphology change associated with *H. volcanii* motile rod differentiation is known to be dependent on CetZ1. The authors effectively state the opposite, so please correct this. Again, always specify what type of differentiation is being described ('morphogenetic' is not specific enough to define the type described here, and would certainly cause confusion for readers in future).

Response: We are very appreciated for this comment. The mistake has been corrected according to the suggestion.

6. There is also no logical basis given for the claim that rod shape may be incompatible with the observed specific type of morphogenesis/cellular differentiation (not pleomorphism generally) in the current species. The authors do not see rod development in their species described here, but that does not mean they cannot form rods under other conditions. I would recommend keeping these statements more open to potential discoveries to the contrary in future.

Response: This sentence has been revised accordingly.

Reviewer #2 (Remarks to the Author):

This is exciting work, and in this revised iteration the authors have done a great job of addressing the comments provided by the previous reviewers. I had only relatively minor comments for the authors to consider.

1) Lines 112-113 and Figure 1b: Spore chains are beautifully evident from the solid culture samples, but they are not as obvious from the liquid-grown culture

Response: Figure 1b has been replaced by another one which could more clearly show the spores in liquid culture.

2) Extended figure 1b – growth (presumably lawns?) is not all that obvious on many of the plates grown at lower temperatures

Response: The color of the mycelium is very close to the color of the medium, so there is no effective color contrast, and the fact that the picture was reduced in size makes the growth not obvious to observe. We have edited the figures from scratch and tried our best to make them legible.

3) Supplementary plate pictures – it is often challenging to see growth relative to the background plate (Figures 1,2)

Response: The haloarchaea do not grow well on the relevant media, together with the white or yellow color of the mycelia close to the media, which makes them more difficult to show in figures. We have edited the figures from scratch and tried our best to make them legible.

4) Lines 253-254: It is not clear why 'immunity' is relevant and connected with the morphogenic characteristics being described in this section?

Response: Thanks for your advice! We deleted this description.

5) Line 315: it isn't clear why it is being proposed that the oligopeptide permease may only be involved in submerged sporulation (and not also during growth on solid substrates)? Was bialaphos resistance tested only during liquid culture? – Figure 5d suggests it was done on solid plates. There is no information about how this testing was done in the methods section of the manuscript – it would be helpful to include this information.

Response: The function of operon *bldKA-KE* has been characterized in *Streptomyces coelicolor*¹ and *Streptomyces griseus*². The deletion of *bldKA-KE* caused a bald phenotype that has only substrate mycelium. Therefore, an oligopeptide permease encoded by *bldKA-KE* should be responsible for the import of an extracellular signal governing aerial mycelium formation. The bialaphos resistance was only found in solid plate culture. The information about this test has been added in methods section.

Related references:

[1] Nodwell, J. R., McGovern, K., & Losick, R. (1996). An oligopeptide permease responsible for the import of an extracellular signal governing aerial mycelium formation in *Streptomyces coelicolor*. *Molecular microbiology*, 22(5), 881-893.

[2] Akanuma, G., Ueki, M., Ishizuka, M., Ohnishi, Y., & Horinouchi, S. (2011). Control of aerial mycelium formation by the BldK oligopeptide ABC transporter in *Streptomyces griseus*. *FEMS microbiology letters*, 315(1), 54-62.

6) Lines 324-336: For this section, actinobacteria and cyanobacteria are highlighted, but then all examples discussed are from *Bacillus subtilis*. It may be worth including the sporulating Firmicutes in the opening sentence, and then

also making clear that the YlbF, BolA and AbrB-family regulators have all been characterized in *Bacillus subtilis*.

Response: The opening sentence has been revised according to this suggestion.

7) Line 664: This would be a negative control, not a positive control.

Response: These descriptions were the method of S-layer protein identification based on gel-separation strategy. The gel band was cut based on protein molecular weight and abundance.

8) Do the transitional mutants shown in Fig. 5c grow better than wild type or the bald mutants? Growth seems to be much more dense for this mutant than any of the other strains, both in the presence and absence of bialaphos. This could be worth commenting on.

Response: Indeed, this is a noteworthy phenomenon. However, when we conducted this test, the amount of inoculum is very difficult to control (because mutants did not produce spore), so it seems that the difference in growth may be related to the amount of inoculum. In future work we will further explore the possible differences in growth of mutant phenotypes.

Reviewer #3 (Remarks to the Author):

1) While the manuscript is greatly improved, the presentation would benefit from careful copyediting and expanding more details in the methods section. Suggestions are included on the attached pdf to help in the revision.

Response: We are very appreciated for the detailed correction comments and suggestions. We have revised accordingly in the new version of our manuscript.

2) In general, the reader should be able to understand the figures from the legend without having to refer to the methods and results sections.

The methods should be described in enough detail to all another lab to reproduce them, especially for those involving the biological materials, ie. isolation of spores, etc.

Response: We have added more information in our methodology part as suggested.

3) Extended Data Fig. 8 | The isolation and characteristics of YIM 93972 mutants panels c and d. What does it mean when the mutation includes the entire coding region, ie. such as T4, last column.

Response: The genome of the mutant was re-sequenced by using the Illumina platform. Mapping all raw reads to reference (complete genome sequence) and counting the frequency of the mutated base could demonstrate that the mutation was reliable and not caused by sequencing error. 0|242 means that there were 242 reads with mutated base (T), but no reads with wild-type base (C).

4) Supplementary Fig. 3 | The analysis of quantitative proteomics by TMT-labeling. The point of c is not clear. What do the fraction numbers correspond to?

Response: The labeled peptide mixtures were fractionated by RP HPLC with a 60 min linear gradient. Fraction was collected every 1 min (Supplementary Fig. 3b). The fraction number was named in turn, and each represents an eluting fraction. The Fraction number was added in revised Fig. 3b.